# OsNF-YB7 inactivates OsGLK1 to inhibit chlorophyll biosynthesis in rice embryo

Zongju Yang[1,2†], Tianqi Bai[1†], Zhiguo E[3†], Baixiao Niu[1,2], Chen Chen[1,2]*

[1]Jiangsu Key Laboratory of Crop Genomics and Molecular Breeding/ Zhongshan Biological Breeding Laboratory, Agricultural College of Yangzhou University, Yangzhou, China; [2]Jiangsu Co-Innovation Center for Modern Production Technology of Grain Crops/ Key Laboratory of Plant Functional Genomics of the Ministry of Education, Agricultural College of Yangzhou University, Yangzhou, China; [3]State Key Laboratory of Rice Biology and Breeding, China National Rice Research Institute, Hangzhou, China

*For correspondence:
chenchen@yzu.edu.cn

†These authors contributed equally to this work

Competing interest: The authors declare that no competing interests exist.

**Abstract** As a master regulator of seed development, Leafy Cotyledon 1 (LEC1) promotes chlorophyll (Chl) biosynthesis in *Arabidopsis*, but the mechanism underlying this remains poorly understood. Here, we found that loss of function of *OsNF-YB7*, a *LEC1* homolog of rice, leads to chlorophyllous embryo, indicating that *OsNF-YB7* plays an opposite role in Chl biosynthesis in rice compared with that in *Arabidopsis*. OsNF-YB7 regulates the expression of a group of genes responsible for Chl biosynthesis and photosynthesis by directly binding to their promoters. In addition, OsNF-YB7 interacts with Golden 2-Like 1 (OsGLK1) to inhibit the transactivation activity of OsGLK1, a key regulator of Chl biosynthesis. Moreover, OsNF-YB7 can directly repress *OsGLK1* expression by recognizing its promoter in vivo, indicating the involvement of OsNF-YB7 in multiple regulatory layers of Chl biosynthesis in rice embryo. We propose that OsNF-YB7 functions as a transcriptional repressor to regulate Chl biosynthesis in rice embryo.

## eLife assessment

This paper provides **important** insights into the role of rice OsNF-YB7, an ortholog of Arabidopsis LEC1, in chlorophyll biosynthesis, uncovering the genetic and molecular basis for negative regulation of chlorophyll production in the rice embryo. Mutational analysis, gene expression profiles and protein interaction combine for **convincing** evidence that OsNF-YB7 represses chlorophyll biosynthesis.

## Introduction

Angiosperms can be divided into chloroembryophytes and leucoembryophytes, depending on the presence or absence of chlorophyll (Chl) in the embryo, respectively (**Puthur et al., 2013**; **Smolikova and Medvedev, 2016**). Some plant species such as *Arabidopsis* (*A. thaliana*) produce chloroembryos, which have photochemically active chloroplasts capable of producing assimilates that are further converted into reserve biopolymers (**Simkin et al., 2020**). Grass species, such as rice (*Oryza sativa*), are unable to produce Chl during embryo development. To the best of our knowledge, it remains completely unclear what determines the ability to biosynthesize Chl in plant embryo. However, studies have identified several genes that contribute to Chl degradation for chloroembryos, the mutation of which can lead to a stay-green phenotype in mature seeds such as Mendel's green pea (**Sato et al., 2007**; **Delmas et al., 2013**; **Wang et al., 2018**; **Li et al., 2017**; **Thomas and Ougham, 2014**).

*Golden 2-Like* (*GLK*) genes encode GARP-type transcription factors (TFs), which are key components regulating chloroplast development and Chl biosynthesis in plants (*Rossini et al., 2001*; *Fitter et al., 2002*; *Nakamura et al., 2009*). GLK can recognize the CCAATC *cis*-element of the Chl biosynthesis- and photosynthesis-associated nuclear genes to trigger their expression (*Waters et al., 2009*). The G-box (CACGTG) is also enriched in the GLK-targeted genes in *Arabidopsis*, possibly due to GLK being able to interact with certain G-box binding factors (*Tamai et al., 2002*). A genetic study showed that GLK activates Chl biosynthesis in roots, in a manner dependent on the G-box binding TF Elongated Hypocotyl 5 (HY5) (*Kobayashi et al., 2012*).

There are two copies of *GLK* in the rice genome, designated as *OsGLK1* and *OsGLK2* (*Fitter et al., 2002*). They redundantly regulate a set of genes, such as rice *Chlorophyllide A Oxygenase* (*OsCAO*) and *Protochlorophyllide Oxidoreductase A* (*OsPORA*) responsible for Chl biosynthesis, and *Light Harvesting Complex B1* (*OsLHCB1*) and *OsLHCB4* responsible for photosynthesis (*Nakamura et al., 2009*; *Wang et al., 2013*; *Sakuraba et al., 2017*). *OsGLK1* overexpression in rice leads to green calli and chloroplast development in the vascular bundles (*Nakamura et al., 2009*; *Wang et al., 2017*). Organelle development in rice vascular sheath cells is induced by ectopically expressed maize (*Zea mays*) *GLK* genes, mimicking a key step in the evolutionary transition from C3 to C4 plants (*Wang et al., 2017*). In accordance with this, the ectopic expression of maize *GLK*s in rice can boost biomass and grain yield by facilitating chloroplast development and photosynthesis (*Li et al., 2020*; *Yeh et al., 2022*). Moreover, the overexpression of maize *GLK*s in calli was shown to improve the ability of rice and maize to regenerate (*Luo et al., 2023*). A recent study found that *OsGLK1* also participates in tapetum plastid development and programmed cell death, consequently affecting pollen fertility in rice (*Zheng et al., 2022*). The findings indicate that GLK plays multiple roles in relation of plant development.

Leafy Cotyledon 1 (LEC1), a member of the nuclear factor Y (NF-Y) TF family, is a central regulator controlling many aspects of seed development, such as Chl accumulation in the embryo (*Meinke et al., 1994*; *Pelletier et al., 2017*). LEC1 can also act as a pioneer TF to regulate flowering by reprogramming the embryonic chromatin state (*Tao et al., 2017*). Previous studies have reported that *lec1* mutants have paler green embryos than wild-type (WT) at maturation in *Arabidopsis* (*Meinke et al., 1994*; *West et al., 1994*). LEC1 transcriptionally regulates the expression of genes that encode light-reaction components of photosystems I and II, as well as the expression of genes involved in chloroplast biogenesis in *Arabidopsis* and soybean embryos (*Pelletier et al., 2017*; *Jo et al., 2020*). These findings suggest that LEC1 is important for photosynthesis and chloroplast development during seed development. However, the underlying molecular mechanisms remain largely unclear.

It is still a mystery why plant species such as rice cannot synthesize chlorophyll in embryos, while species such as *Arabidopsis* can. There are two LEC1 homologs, *OsNF-YB7* and *OsNF-YB9*, encoded by the rice genome (*Zhiguo et al., 2018*). *OsNF-YB7* is restricted to the embryo, and defective *OsNF-YB7* may result in seed lethality (*Niu et al., 2021b*; *Zhang and Xue, 2013*). Here, we found that OsNF-YB7 acts as a key inhibitor of Chl biosynthesis in rice embryo. It inactivates the transactivation activity of OsGLK1, at multiple regulatory layers, to inhibit Chl accumulation in the embryo of rice, explaining the achlorophyllous embryo produced in rice.

## Results

### Loss of function of *OsNF-YB7* leads to chloroembryos

By observing seeds produced by the loss-of-function mutant of *OsNF-YB7*, we surprisingly found that the *osnf-yb7* embryo was greenish (*Figure 1A and B*), suggesting that *OsNF-YB7* plays a negative role in chloroplast biogenesis or Chl biosynthesis, or both, during embryogenesis. We therefore examined the phenotype of green embryos at various seed development stages. The results showed that WT had an achlorophyllous embryo throughout embryonic development, whereas *osnf-yb7* embryo turned green at 5 days after fertilization (DAF) and the chloroembryo remained green until maturity (*Figure 1—figure supplement 1A–C*). In support of this, transmission electron microscopy (TEM) showed well-developed chloroplast in the scutellum tip of the mutant, but this was not seen in the WT sections (*Figure 1C*). By measuring Chl contents in the developing and mature embryos, we found that total Chl content in *osnf-yb7* was consistently higher than that in the WT, both at 10 DAF and at maturation (30 DAF) (*Figure 1D*). As revealed by confocal laser scanning microscopy (CLSM), Chl

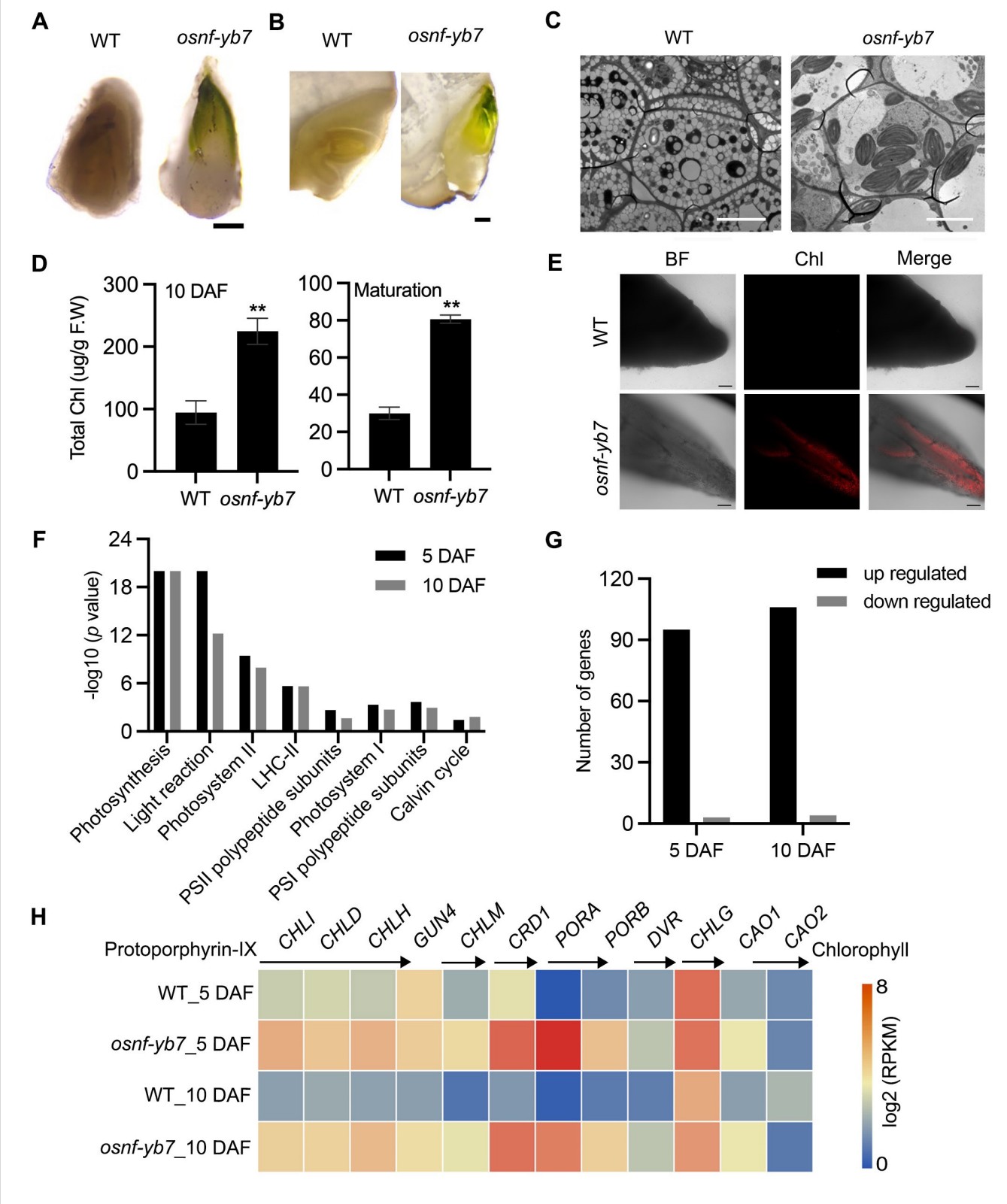

**Figure 1.** OsNF-YB7 negatively regulates Chlorophyll (Chl) biosynthesis in embryo. (**A, B**) Morphologies of wild-type (WT) and *osnf-yb7* detached embryos at 10 days after fertilization (DAF) (**A**) and longitudinally dissected embryos at maturation (**B**). Scale bars = 200 μm. (**C**) Transmission electron microscopy images of embryos from WT and *osnf-yb7* at 10 DAF. Scale bars = 5 μm. (**D**) Chl levels in WT and *osnf-yb7* embryos at 10 DAF and maturation. Data are means ± SD (n=3). **, p<0.01; Student's *t*-test was used for statistical analysis. (**E**) Chl autofluorescence of WT and mutant embryos

*Figure 1 continued on next page*

*Figure 1 continued*

at 5 DAF. BF, bright field; Chl, Chl autofluorescence. Scale bars = 100 µm. (**F**) Photosynthesis-related pathways enriched among the differentially expressed genes (DEGs) identified from the 5- and 10-DAF-old *osnf-yb7* embryos compared to WT, respectively. (**G**) Most of the photosynthesis-related genes were upregulated in the mutant embryos. (**H**) A heat map shows the expression of the Chl biosynthesis-related genes in the WT and *osnf-yb7* embryos at 5 and 10 DAF. Reads per kilobase per million mapped reads (RPKM) is used to indicate the expression level.

The online version of this article includes the following figure supplement(s) for figure 1:

**Figure supplement 1.** Mutation of *OsNF-YB7* leads to chloroembryo.

**Figure supplement 2.** Light is required but not sufficient for Chl biosynthesis in the chloroembryo of *osnf-yb7*.

**Figure supplement 3.** Mutation of *OsNF-YB7* actives the expression of photosynthesis-related genes.

**Figure supplement 4.** OsNF-YB7 negatively regulates the expression of light harvest and Chl biosynthesis-associated genes.

autofluorescence was detectable as early as 5 DAF in *osnf-yb7*, indicating the initiation of Chl accumulation at this stage (***Figure 1E***). This is consistent with our previous finding that *OsNF-YB7* was highly activated at 5 DAF in rice embryo (***Niu et al., 2021b***).

Light is a critical signal triggering Chl biosynthesis (***Wietrzynski and Engel, 2021***). To determine whether the Chl biosynthesis in *osnf-yb7* is induced by light, we investigated the embryo phenotype of WT and *osnf-yb7* in the dark, using aluminum foil to cover rice panicles prior to flowering. The result showed that, similar to WT, *osnf-yb7* embryo was achlorophyllous in the dark, although the embryogenesis defects, such as degenerated epiblast and coleorhiza, and maldeveloped coleoptile, were still observable (***Figure 1—figure supplement 2A–H***). This indicated that Chl biosynthesis in the mutant is light dependent. Only a small amount of light can be perceived by rice embryos because the external hulls block light penetration (***Simkin et al., 2020***). We removed the hulls to directly expose the embryo to light; however, Chl accumulation still failed to occur in the WT embryo (***Figure 1—figure supplement 2I–L***), suggesting that there are internal signals that repress Chl biosynthesis in rice embryo. We suspected that the *OsNF-YB7* mutation may attenuate the activity of such inhibitors; alternatively, OsNF-YB7 itself could be an inhibitor.

## OsNF-YB7 negatively regulates photosynthesis- and Chl biosynthesis-related genes

Using 5- and 10-DAF-old WT and *osnf-yb7* embryos, we previously performed deep sequencing of the transcriptome (RNA-seq) to identify possible downstream genes of OsNF-YB7 (***Niu et al., 2021a***). As revealed by Mapman analysis, photosynthesis-related pathways, such as photosystem I, photosystem II, and the light reaction, were significantly enriched for the differentially expressed genes (DEGs) in the mutant embryos (***Figure 1F***, ***Figure 1—figure supplement 3A and B***). Moreover, 96.9% (95/98) and 96.4% (106/110) of the photosynthesis-related DEGs that identified from 5- and 10–DAF-old embryos, respectively, were upregulated in the mutant (***Figure 1G***). To confirm this finding, we next examined the expression of *OsLHCA*s and *OsLHCB*s, which are primarily associated with photosystems I and II, respectively, via quantitative real-time PCR (RT-qPCR). It showed higher expression of all of the studied *OsLHCA*s and *OsLHCB*s in the *osnf-yb7* embryo at 10 DAF (***Figure 1—figure supplement 4A***). Likewise, many of the genes participating in Chl biosynthesis, including rice *Genomes Uncoupled 4* (*OsGUN4*), *Mg-Chelatase H Subunit* (*OsCHLH*), *OsCHLI*, *OsCHLD*, *Copper Response Defect 1* (*OsCRD1*), *OsPORA*, *OsPORB*, and *Divinyl Reductase* (*OsDVR*), were significantly activated in the mutant (***Figure 1H*** and ***Figure 1—figure supplement 4B***). The findings indicated that OsNF-YB7 might act as a repressor of Chl biosynthesis and photosynthesis, which is opposite to the role of its homologue LEC1 in *Arabidopsis*.

Since OsNF-YB7 is a TF, we assumed that it may directly regulate the expression of genes related to Chl biosynthesis and photosynthesis, such as *OsPORA*, and *OsLHCB4*, which were significantly activated in the *osnf-yb7* embryo (***Figure 2A and B***). We first generated transgenic lines that overexpressed *OsNF-YB7*, tagged with either green fluorescent protein (*NF-YB7-GFP*) or 3×Flag (*NF-YB7-Flag*) in the Zhonghua11 (ZH11, *O. sativa* ssp. *geng/japonica*) or Kitaake (*O. sativa* ssp. *geng/japonica*) background, respectively. As expected, *OsPORA* and *OsLHCB4* were significantly downregulated in leaves of the *OsNF-YB7* overexpressors (***Figure 2C and D***, and ***Figure 1—figure supplement 4C***). Similar to previously reported (***Zhang and Xue, 2013***; ***Ito et al., 2011***), the OsNF-YB7 overexpression lines displayed severe reproductive development defects, which prevented us from obtaining

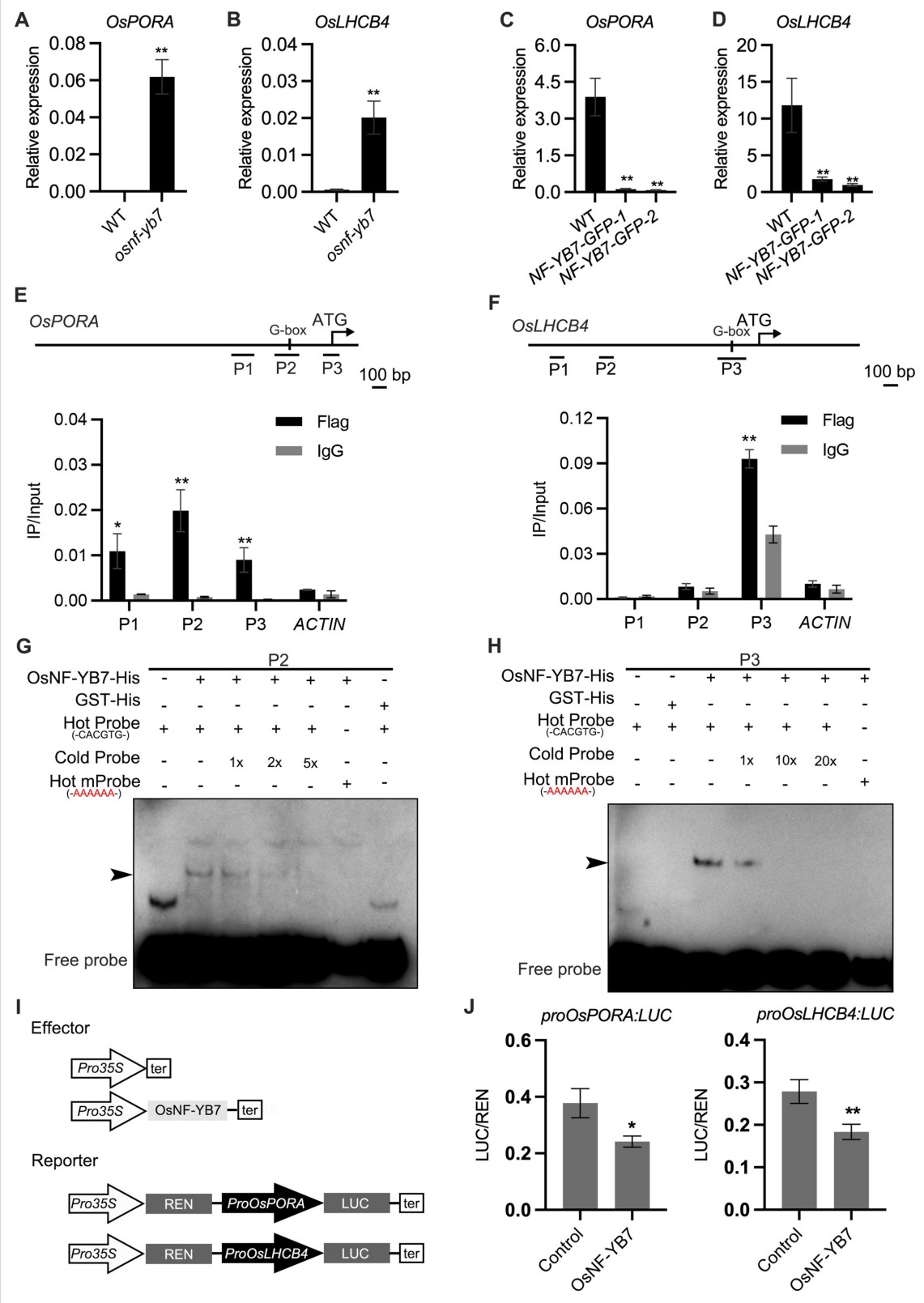

**Figure 2.** OsNF-YB7 binds to the promoters of *OsPORA* and *OsLHCB4* to regulate their expression. (**A, B**) Quantitative real-time PCR (RT-qPCR) analysis of the transcription levels of *OsPORA* (**A**) and *OsLHCB4* (**B**) in the embryos of WT and *osnf-yb7* at 10 DAF. Data are means ± SD (n=3). **, p<0.01; Student's *t*-test was used for statistical analysis. (**C, D**) Expression of *OsPORA* (**C**) and *OsLHCB4* (**D**) in the leaves of WT and *OsNF-YB7*-overexpressing transgenic plants (*NF-YB7-GFP*). Data are means ± SD (n=3). **, p<0.01; Student's *t*-test was used for statistical analysis. (**E, F**) Chromatin

*Figure 2 continued on next page*

*Figure 2 continued*

immunoprecipitation assay coupled with quantitative PCR (ChIP-qPCR) analyses showing the enrichment of OsNF-YB7 at the *OsPORA* (**E**) and *OsLHCB4* (**F**) promoters in 14-day-old *OsNF-YB7-Flag* seedlings. Precipitated DNA was quantified by qPCR and DNA enrichment is displayed as a percentage of input DNA. Data are means ± SD (n=3). *, $p<0.05$; **, $p<0.01$; Student's *t*-test was used for statistical analysis. *ACTIN* was used as a nonspecific target gene. Diagrams in the upper panel showing the promoter structures of *OsPORA* and *OsLHCB4*, and the PCR amplicons used for ChIP-qPCR. (**G, H**) Electrophoretic mobility-shift assays (EMSAs) showing that OsNF-YB7 directly binds to the promoters of *OsPORA* (**G**) and *OsLHCB4* (**H**). Hot probes were biotin-labeled. The hot mProbes contain mutant nucleic acid from CACATG to AAAAAA. The arrow heads indicate the shift bands. (**I**) Schematic diagram displaying the constructs used in the dual luciferase reporter (DLR) assays of **J**. LUC, firefly luciferase; REN, *Renilla* luciferase. (**J**) DLR assays showing that OsNF-YB7 directly represses the promoter activities of *OsPORA* and *OsLHCB4*. Data are means ± SD (n=3). *, $p<0.05$; **, $p<0.01$; Student's *t*-test was used for statistical analysis.

The online version of this article includes the following source data for figure 2:

**Source data 1.** Uncropped and labelled gels.

**Source data 2.** Raw unedited gels.

sufficient embryo tissues for subsequent experiments. Instead, using the *NF-YB7-Flag* seedling, we conducted a chromatin immunoprecipitation assay coupled with quantitative PCR (ChIP-qPCR). The results showed that OsNF-YB7 was highly enriched in the promoter regions of *OsPORA* and *OsLHCB4* harboring the G-box motif (***Figure 2E and F***), a putative binding site of OsNF-YB7 (***Guo et al., 2022***). To confirm the ability of OsNF-YB7 to bind to the *OsPORA* and *OsLHCB4* promoters, we next performed electrophoretic mobility-shift assays (EMSAs), using recombinant OsNF-YB7-His protein, and biotin-labeled subfragments of the *OsPORA* or *OsLHCB4* promoters containing G-boxes. The results showed that OsNF-YB7-His was able to bind to the labeled probes, and the shifted band signals were substantially weakened upon application of the unlabeled cold probes or hot probes with a mutated G-box (***Figure 2G and H***).

To confirm that the binding of OsNF-YB7 represses target gene expression, dual luciferase (LUC) reporter (DLR) assays were performed. We first generated LUC reporters driven by the *OsPORA* or *OsLHCB4* promoters (designated as *proOsPORA:LUC* and *proOsLHCB4:LUC* hereafter) (***Figure 2I***). When the reporters were coexpressed with OsNF-YB7, which acted as an effector, in rice protoplasts, we found that OsNF-YB7 significantly repressed the activity of both *proOsPORA:LUC* and *proOsLHCB4:LUC* (***Figure 2J***). Taken together, these results suggested that OsNF-YB7 directly binds to the promoters of photosynthesis- and Chl-biosynthesis-related genes and represses their transcription.

## OsNF-YB7 represses *OsGLK1* in the embryo

Several TFs that regulate Chl biosynthesis or chloroplast development have been identified in plants (***Jarvis and López-Juez, 2013***). Some of these, such as rice *OsGLK1*, HY5-like 1 (*OsHY5L1*), *PIF-like 14* (*OsPIL14*), and *GATA Nitrate-inducible Carbon-metabolism-involved* (*OsGNC*), were found to be upregulated in the embryo of *osnf-yb7* (***Figure 3A***).

*OsGLK1* was the most strikingly activated TF (***Figure 3A and B***), while it was significantly repressed in *NF-YB7-GFP* (***Figure 3C***). To test the idea that *OsGLK1* is a direct downstream target of OsNF-YB7, a ChIP-qPCR assay was first carried out using *NF-YB7-Flag* transgenic seedlings. The results showed that the promoter segments P3 and P4 were significantly enriched in the immunoprecipitated chromatin (***Figure 3D***), suggesting that OsNF-YB7 was able to bind to the *OsGLK1* promoter in vivo. However, we failed to validate the binding ability in vitro using the EMSA assay (***Figure 3—figure supplement 1***), suggesting that OsNF-YB7 requires other NF-Ys or TFs to form a TF complex to recognize the promoter, as many NF-Y members do for function (***Laloum et al., 2013***). A DLR assay was next performed to investigate the negative regulation of OsNF-YB7 on *OsGLK1* transcription using rice protoplasts. The *OsGLK1* promoter was inserted upstream of the 5×upstream activating sequence (UAS) as a reporter; OsNF-YB7 was fused C-terminally to the DNA binding domain (BD) of the yeast GAL4 and the herpes virus VP16 transactivation domain (VP16:OsNF-YB7), as an effector (***Figure 3E***). Coexpression of VP16:OsNF-YB7 with the reporter in rice protoplasts significantly decreased the transcriptional activity of VP16 (***Figure 3F***), indicating that OsNF-YB7 represses *OsGLK1* promoter activity.

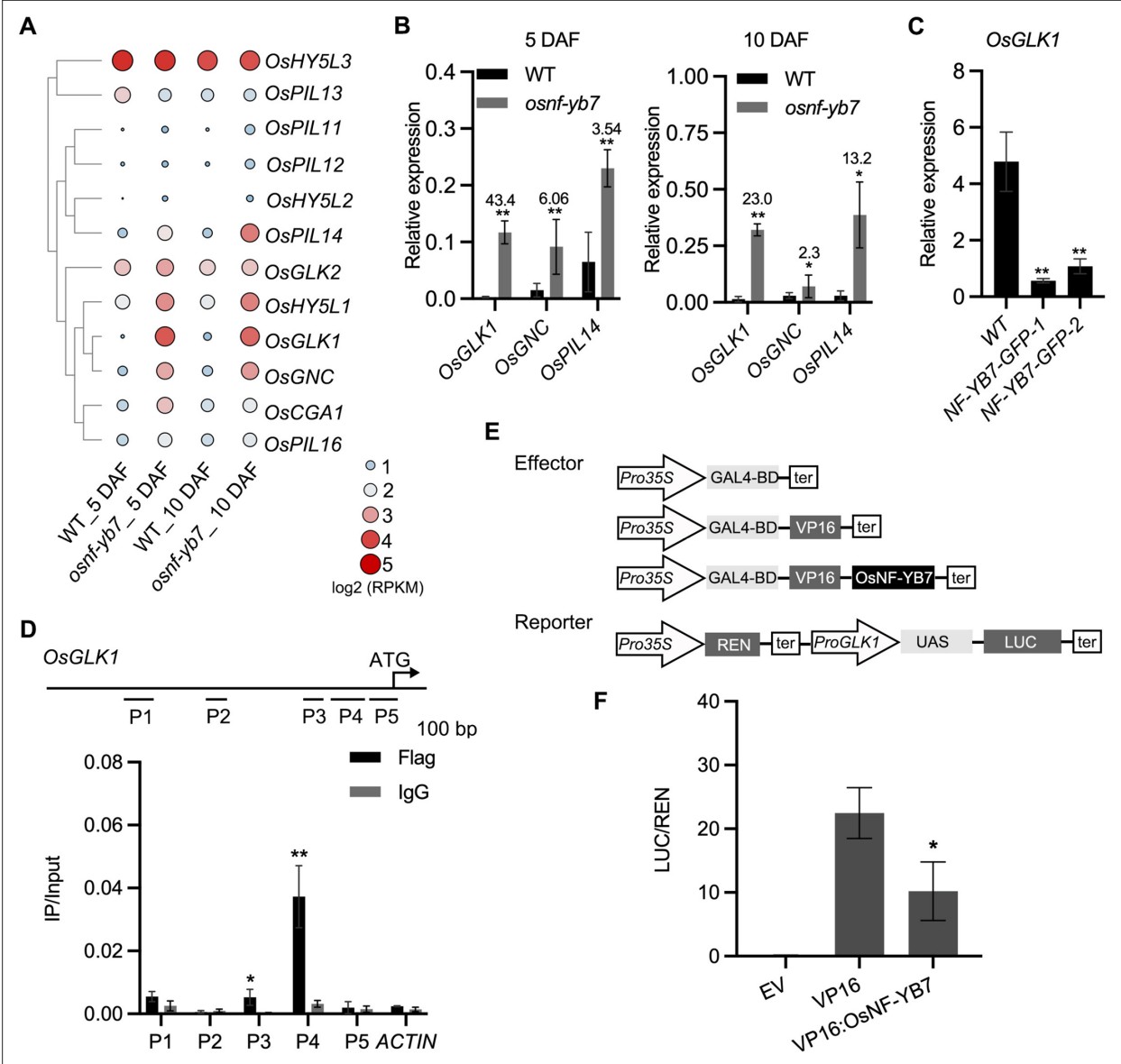

**Figure 3.** OsNF-YB7 associates with the promoter of *OsGLK1* and represses its expression. (**A**) A heat map showing the expression of transcription factors associated with Chl biosynthesis and chloroplast development in the 5- and 10-DAF-old embryos of WT and *osnf-yb7*. The colored dots indicate log$_2$(RPKM mean) of the genes in three biological replicates. (**B**) RT-qPCR analysis of *OsGLK1*, *OsGNC*, and *OsPIL14* expression levels in the embryos from WT and *osnf-yb7* at 5- and 10 DAF. Numbers represent fold changes of expression. Data are means ± SD (n=3). *, $p<0.05$; **, $p<0.01$; Student's *t*-test was used for statistical analysis. (**C**) RT-qPCR analysis of *OsGLK1* expression levels in leaves from WT and *NF-YB7-GFP*. Data are means ± SD (n=3). **, $p<0.01$; Student's *t*-test was used for statistical analysis. (**D**) ChIP-qPCR analysis showing the enrichment of OsNF-YB7 at the *OsGLK1* promoter in 14-day-old *OsNF-YB7-Flag* seedling. Chromatin of each sample was immunoprecipitated using anti-Flag or IgG antibodies. Precipitated DNA was quantified by qPCR and DNA enrichment is displayed as a percentage of input DNA. Data are means ± SD (n=3). *, $p<0.05$; **, $p<0.01$; Student's *t*-test was used for statistical analysis. *ACTIN* was used as a nonspecific target gene. The experiment was performed three times with similar results. The diagram in the upper panel showing the promoter structure of *OsGLK1* and PCR amplicons (P1, P2, P3, P4, and P5) used for ChIP-qPCR. (**E**) Schematic diagram displaying the constructs used in the DLR assays of **F**. LUC, firefly luciferase; REN, *Renilla* luciferase; UAS, upstream activating sequence. (**F**) DLR assays showing that OsNF-YB7 represses the promoter activity of *OsGLK1*. Data are means ± SD (n=3). *, $p<0.05$; Student's *t*-test was used for statistical analysis.

The online version of this article includes the following source data and figure supplement(s) for figure 3:

**Figure supplement 1.** OsNF-YB7 does not directly binds to the promoter of *OsGLK1* in vitro.

**Figure supplement 1—source data 1.** Uncropped and labelled gels.

**Figure supplement 1—source data 2.** Raw unedited gels.

## OsGLK1 is involved in OsNF-YB7-regulated Chl biosynthesis in embryo

To confirm the contribution of *OsGLK1* for the production of chloroembryo in *osnf-yb7*, we first generated *OsGLK1*-overexpressing lines (Os*GLK1-OX*) driven by the rice ubiquitin promoter. In association with over-accumulated Chl in the glume and seed coat of the transformant, RT-qPCR and Western blot assays confirmed *OsGLK1* activation in *OsGLK1-OX* (*Figure 4—figure supplement 1A-C*). As observed in *osnf-yb7*, green embryos were produced (*Figure 4A*). The Chl content in *OsGLK-OX* were higher than that in WT (*Figure 4B and C*), suggesting that *OsGLK1* overexpression in rice embryo induces Chl biosynthesis.

Using a clustered regularly interspaced short palindromic repeats (CRISPR)/CRISPR-associated 9 (Cas9) cassette including three tandemly arrayed guide RNAs targeting *OsNF-YB7*, *OsGLK1*, and *OsGLK2*, respectively, we successfully obtained the *osnf-yb7;osglk2* double mutant and the *osnf-yb7;osglk1;osglk2* triple mutant (*Figure 4—figure supplement 2A and B*). By analyzing the embryos that the mutants produced, we found that significantly less Chl accumulated in *osnf-yb7;osglk1;osglk2* than in *osnf-yb7* and *osnf-yb7;osglk2* (*Figure 4D–G* and *Figure 4—figure supplement 3*). In comparison to the achlorophyllous embryo of WT, the *osnf-yb7;osglk1;osglk2* triple mutant still showed somewhat green coloration in the apical part of the embryos (*Figure 4G*). We hypothesized that this was at least partially due to that genes like *OsLHCB4* and *OsPORA* can be induced by the mutation of *OsNF-YB7*, given the fact that OsNF-YB7 represses the genes' expression independent of OsGLK1 (*Figure 2J*).

## OsNF-YB7 and OsGLK1 regulate a common set of genes in the embryo

We next performed RNA-seq analysis to explore the transcriptomic changes in the chloroembryo of *OsGLK1-OX* at 10 DAF. More than 64.4% of the DEGs identified in *OsGLK1-OX* were overlapped with the ones identified in *osnf-yb7*, the vast majority of which were either activated (60.6%) or repressed (27.6%) in both *OsGLK1-OX* and *osnf-yb7* (*Figure 4H and I*, and *Figure 4—source data 1 and 2*). As revealed by the Gene Ontology (GO) analysis, genes involved in Chl biosynthesis and photosynthesis, such as *OsPORA* and *OsLHCB4*, were markedly enriched among the common DEGs (*Figure 4J* and *Figure 4—figure supplement 4*), implying that OsNF-YB7 and OsGLK1 antagonistically regulate a common set of genes for Chl biosynthesis and photosynthesis.

The ChIP-qPCR results showed that OsGLK1 associated with the regions of the *OsPORA* and *OsLHCB4* promoters to which OsNF-YB7 binds (*Figure 5A and B*). In agreement with this, the EMSA results suggested that OsGLK1 directly binds to the same DNA probes of *OsPORA* and *OsLHCB4* in vitro (*Figure 5C and D*). In opposite to OsNF-YB7, when we coexpressed the OsGLK1 effector vector with the reporter vector *proOsLHCB4:LUC* or *proOsPORA:LUC* in rice protoplasts, OsGLK1 showed significant transactivation activity on *OsPORA* and *OsLHCB4* (*Figure 5—figure supplement 1A and B*).

Recently, the putative binding sites of OsNF-YB7 and OsGLK1 were investigated at the whole genome scale (*Guo et al., 2022*; *Tu et al., 2022*). This allows us to test our hypothesis that OsNF-YB7 and OsGLK1 can target a common set of genes involved in Chl biosynthesis and photosynthesis. By reanalyzing the ChIP-seq data, we found that 91.4% (235/257) and 90.7% (167/184) of the OsGLK1-binding genes overlapped with the OsNF-YB7-binding genes in two replicates (*Figure 5E* and *Figure 5—source data 1*), although the peak number of OsNF-YB7 was much greater than that of OsGLK1 (*Guo et al., 2022*; *Tu et al., 2022*). A large number of the common targets were activated in the embryos of *osnf-yb7* and *OsGLK1-OX* at 10 DAF (*Figure 5—figure supplement 2A*). GO analysis suggested that most of the common targets were genes involved in Chl biosynthesis or photosynthesis (*Figure 5—figure supplement 2B*). By retrieving the sequences of the OsNF-YB7 and OsGLK1 binding peaks in the common targets for MEME analysis, we found that the TFs probably recognize similar DNA motifs. A short sequence containing the G-box motif was the most significantly enriched (*Figure 5—figure supplement 3A and B*). There was also enrichment of another sequence containing a CCAAT motif recognized by the NF-Y TF complexes and a CCAATC motif recognized by GLKs, as previously reported (*Waters et al., 2009*; *Pelletier et al., 2017*; *Figure 5—figure supplement 3A and B*). The results suggested that OsNF-YB7 and OsGLK1 can bind to the same region of their common target. In agreement with this, approximately 75.0% and 76.5% of the peak summits of OsGLK1 were located proximally to the summit of OsNF-YB7 in two replicates, within an adjacent region no more than 100 bp away (*Figure 5F* and *Figure 5—source data 1*). Consistent with our

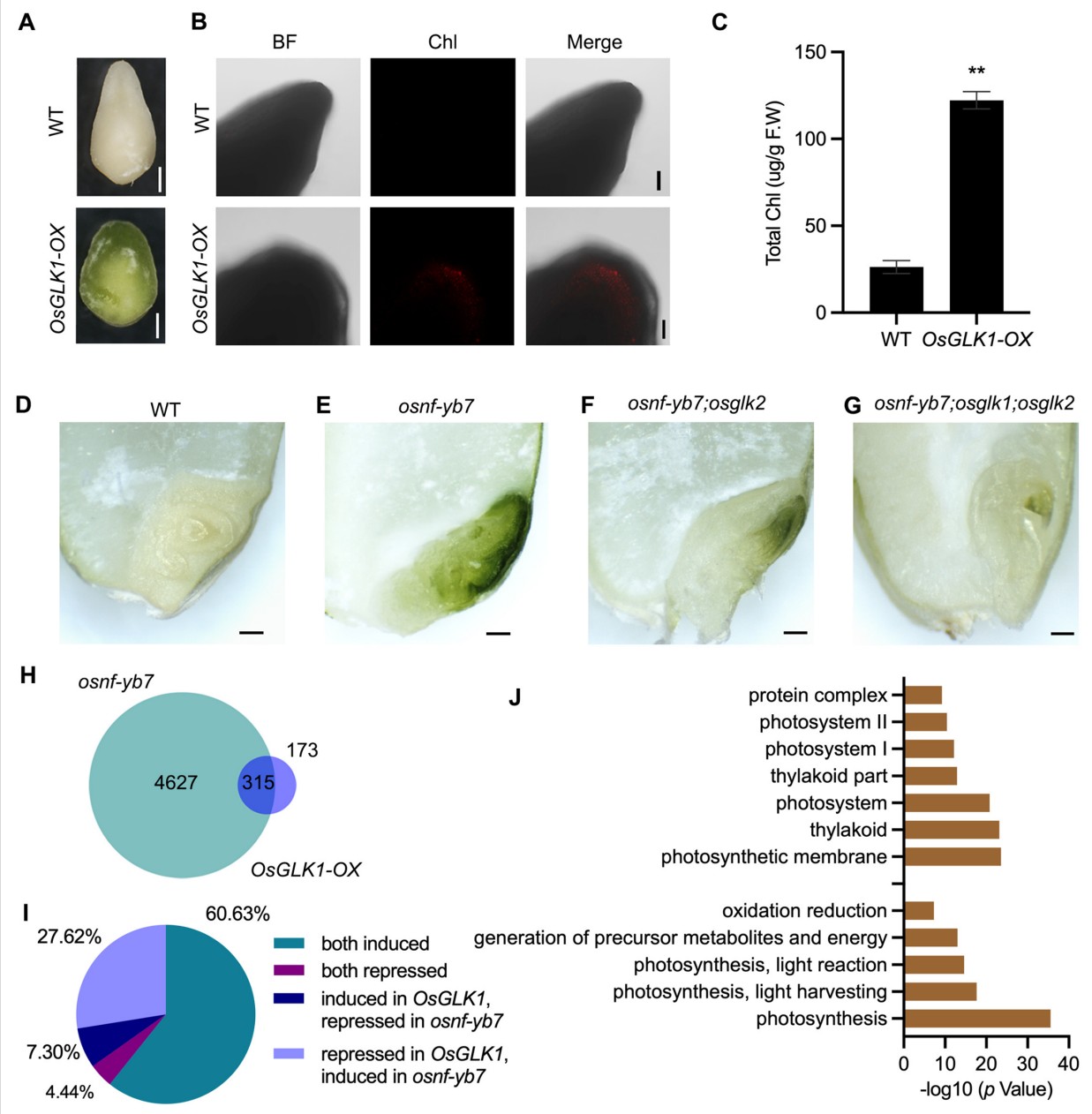

**Figure 4.** Chl biosynthesis in *osnf-yb7* embryo requires active *OsGLK*s. (**A**) Embryo morphologies of WT and Os*GLK1-OX* detached embryos at 10 DAF. Scale bars = 2 mm. (**B**) Chl autofluorescence of the WT and *GLK-OX* embryos at 10 DAF. BF, bright field; Chl, Chl autofluorescence. Scale bars = 100 µm. (**C**) Chl levels in WT and *OsGLK1-OX* embryos at maturation. Data are means ± SD (n=3). **, p<0.01; Student's *t*-test was used for statistical analysis. (**D–G**) Morphologies of the embryos produced by WT (**D**), *osnf-yb7* (**E**), *osnf-yb7;osglk2* double mutant (**F**) and *osnf-yb7;osglk1;osglk2* triple mutant (**G**). Scale bars = 1 mm. (**H**) A Venn diagram showing overlaps of the DEGs identified from the embryos of *osnf-yb7* and *OsGLK1-OX* at 10 DAF. (**I**) A pie chart showing similar transcriptional changes of the common DEGs identified from *osnf-yb7* and *OsGLK1-OX*. (**J**) Gene Ontology (GO) analysis of the common DEGs identified from *osnf-yb7* and *OsGLK1-OX*.

The online version of this article includes the following source data and figure supplement(s) for figure 4:

**Source data 1.** Differentially expressed genes (DEGs) in the 10-DAF-old embryos of *OsGLK1-OX*.

**Source data 2.** Common DEGs identified from the *OsGLK1-OX* and *osnf-yb7* embryos at 10 DAF.

**Figure supplement 1.** Overexpression of *OsGLK1* induces chloroembryo in rice.

**Figure supplement 1—source data 1.** Uncropped and labelled gels.

**Figure supplement 1—source data 2.** Raw unedited gels.

*Figure 4 continued on next page*

**Figure supplement 2.** Generation of *osnf-yb7*, *osglk1*, and *osglk2* high-order mutants.

**Figure supplement 3.** Chl levels in WT, *osnf-yb7* and *osnf-yb7;osglk1;osglk2* embryos at 10 DAF.

biochemical evidence, the ChIP-seq results showed that OsNF-YB7 does bind to the same regions of the *OsPORA* and *OsLHCB4* promoters as OsGLK1 binds to (**Figure 5—figure supplement 3C**).

## OsNF-YB7 physically interacts with OsGLK1 to inhibit its transcriptional activity

To resolve how OsNF-YB7 and OsGLK1 bind to the same regions to regulate their common targets, we speculated that the TFs probably form a dimer in rice in order to exert their functions. We therefore carried out a yeast two-hybrid (Y2H) assay by transforming yeast cells with a bait construct expressing OsNF-YB7 fused with the GAL4 DNA-binding domain (BD), together with a prey construct expressing OsGLK1 fused with the yeast GAL4 activation domain (AD). The results showed that OsNF-YB7 interacts with OsGLK1 in yeast (**Figure 6A and B**). Furthermore, to determine the functional domains required for the interaction, we generated two truncated versions of OsNF-YB7, which contained the N- or C-terminal, and three truncated versions of OsGLK1, which contained the N-terminal, DNA-BD, or GCT box domain, as previously reported (**Fitter et al., 2002**; **Zhang et al., 2021a**; **Figure 6A**). We found that the C-terminus of OsNF-YB7 was sufficient for the interaction (**Figure 6B**). In addition, the GCT box domain of OsGLK1 strongly interacted with the full length or C-terminus of OsNF-YB7, while the DNA-BD of OsGLK1 showed a weak interaction (**Figure 6B**).

A split complementary LUC assay further confirmed the interaction between OsNF-YB7 and OsGLK1 in the epidermal cells of *Nicotiana benthamiana* (**Figure 6C**). As suggested by the bimolecular fluorescence complementation (BiFC) analysis, the interaction occurred exclusively in the nuclei of *N. benthamiana* epidermal cells (**Figure 6D**). Moreover, we transiently coexpressed OsNF-YB7 tagged with GFP (OsNF-YB7-GFP) and OsGLK1 tagged with 3×Flag (OsGLK1-Flag) in rice protoplasts, and co-immunoprecipitation (Co-IP) analysis showed that OsGLK1-Flag could be immunoprecipitated by the anti-GFP antibody (**Figure 6E**), indicating that the interactions do occur in vivo. In addition, both Y2H and split complementary LUC assays showed that OsGLK2 could interact with OsNF-YB7 (**Figure 6—figure supplement 1A and B**). These findings indicated that OsNF-YB7 interacts with OsGLKs, explaining why OsNF-YB7 and OsGLKs share a common set of targets in rice.

To explore the biological meaning of the interaction, we next performed DLR assays using rice protoplasts. Transient expression of OsGLK1 in rice protoplasts substantially activated the reporters, driven by either the *OsPORA* or the *OsLHCB4* promoter; however, when we coexpressed OsNF-YB7 with OsGLK1, the transactivation ability of OsGLK1 was significantly repressed in rice protoplasts (**Figure 6F**). These findings suggested that OsNF-YB7/OsGLK1 dimerization reduces the transactivation ability of OsGLK1 for fine-tuning the Chl biosynthetic and photosynthetic genes, such as *OsPORA* and *OsLHCB4*. The EMSA assay showed that OsGLK1-MBP recombinant proteins could bind to the promoter of *OsPORA*; however, when we incubated the probe with OsGLK1-MBP and OsNF-YB7-His together, the binding ability of OsGLK1 substantially decreased (**Figure 6G**). As a control, when the probe was incubated with OsGLK1-MBP and GST-His, the binding ability of OsGLK1 remained unchanged (**Figure 6G**). These findings implied that the reduced transactivity of OsGLK1 is likely due to the formation of OsNF-YB7/OsGLK1 heterodimers inhibiting the binding of OsGLK1 to its downstream genes.

## Discussion

In crops such as soybean and canola, the presence of green embryos is considered as a valuable trait due to its association with increased photosynthetic capacity, which consequently promotes fatty acid biosynthesis (**Ruuska et al., 2004**). This highlights the potential value of the green embryo. Nevertheless, the degradation of chlorophyll in mature seeds is necessary to prevent adverse effects on seed viability and meal quality (**Chung et al., 2006**).

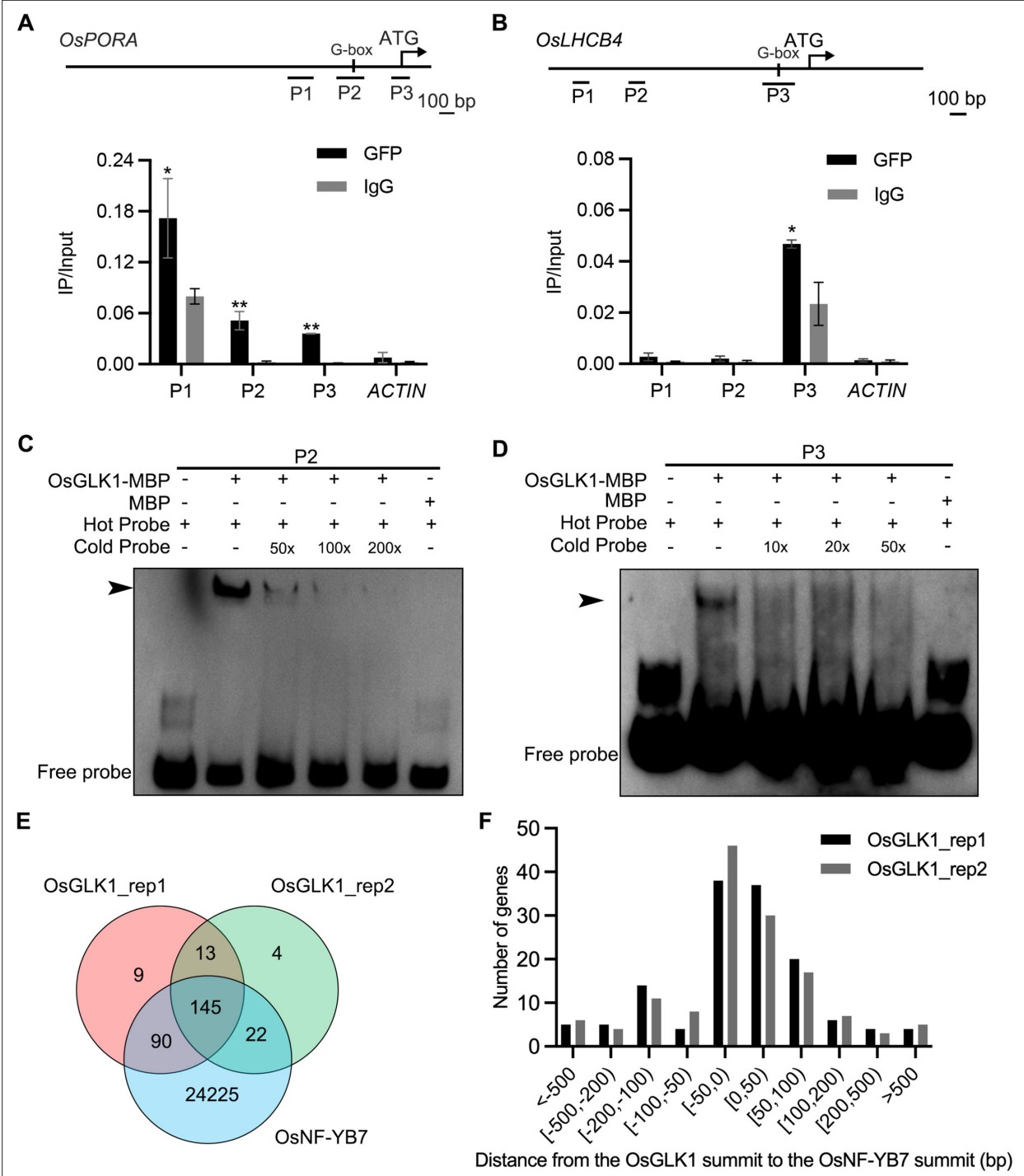

**Figure 5.** OsNF-YB7 and OsGLK1 regulate a common set of genes in the embryo. (**A, B**) ChIP-qPCR analysis showing the enrichment of OsGLK1 on the *OsPORA* (**A**) and *OsLHCB4* (**B**) promoters. PCR amplicons used for ChIP-qPCR are indicated in the schematic diagrams. OsGLK1-GFP was transiently expressed in protoplasts isolated from green tissues of the 14-day-old WT seedling. Chromatin of each sample was immunoprecipitated using anti-GFP or igG antibodies. Precipitated DNA was quantified by qPCR and DNA enrichment is displayed as a percentage of input DNA. Data are means ± SD (n=3). *, p<0.05; **, p<0.01; Student's *t*-test was used for statistical analysis. *ACTIN* was used as a nonspecific target gene. The experiment was performed three times with similar results. (**C, D**) EMSA assays showing that OsGLK1 directly binds to the promoters of *OsPORA* (**C**) and *OsLHCB4* (**D**). The arrow heads indicate the shift bands. (**E**) A Venn diagram showing overlaps of the target genes between OsGLK1 and OsNF-YB7. (**F**) Distribution of the distance between the OsNF-YB7 and OsGLK1 summits showing that OsNF-YB7 and OsGLK1 bind to proximal regions of their common targets.

*Figure 5 continued on next page*

*Figure 5 continued*

The online version of this article includes the following source data and figure supplement(s) for figure 5:

**Source data 1.** Common targets of OsGLK1 and OsNF-YB7.

**Source data 2.** Uncropped and labelled gels.

**Source data 3.** Raw unedited gels.

**Figure supplement 1.** OsGLK1 significantly activates the promoter activities of *OsPORA* and *OsLHCB4*.

**Figure supplement 2.** Many of the OsNF-YB7 and OsGLK1 common targets that activated in *osnf-yb7* and *OsGLK1-OX* are involved in Chl biosynthesis and photosynthesis.

**Figure supplement 3.** OsNF-YB7 and OsGLK1 bind to proximal regions of their common targets.

By surveying 1094 species from 666 genera and 182 families, *Yakovlev and Zhukova, 1980* found that 428 angiosperms produce embryos with the presence of chlorophyll. Seeds with chlorophyllous embryo are scattered throughout the angiosperms and have evolved in many unrelated groups, but they are more prevalent in non-endospermous seeds (*Dahlgren, 1980*). Currently, the evolutionary force driving the divergence of chloroembryophytes and leucoembryophytes remains largely unknown. Embryo photosynthesis contributes a large amount of oxygen to fuel energy-generating pathways in seed (*Simkin et al., 2020*). The presence of chlorophyll in the embryo facilitates photosynthesis at early developmental stages, potentially leading to improved seedling growth and vigor (*Smolikova and Medvedev, 2016*). In many chloroembryophytes, such as *Arabidopsis*, the embryo occupies a large proportion of the seed. From an evolutionary perspective, the presence of chlorophyll in the embryo may promote adaptation in such chloroembryophytes, as more reserves can be accumulated in the seed through active photosynthesis, better supporting the embryo development and subsequent seedling growth (*Sela et al., 2020*). On the other hand, some leucoembryophytes, such as rice, have a persistent endosperm rich in storage reserves to nourish embryo development (*Liu et al., 2022*). The acquisition of the ability to accumulate chlorophyll in the embryo is not necessary in such species. Although chlorophyllous embryos are rare among the 'primitive' angiosperm superorders, they are observed in some (but not all) Nymphaeales species that are surrounded by some amount of endosperm (*Yakovlev and Zhukova, 1980*). As there is a paucity of knowledge regarding the evolution of chlorophyllous embryos, further comprehensive studies are necessary.

Some algae and gymnosperm species have evolved an ability to synthesize Chl in the dark (*Myers, 1940*; *Ranade et al., 2019*; *Bogorad, 1950*). However, it has remained largely unclear whether access to light accounts for the induction of chloroembryos (*Dahlgren, 1980*; *Periasamy and Vivekanandan, 1981*; *Liu et al., 2017*). Here, we found that removing the tissues covering an embryo failed to produce chlorophyllous embryos in the WT; however, light avoidance inhibited Chl accumulation in *osnf-yb7* embryos (*Figure 1—figure supplement 2A–H*). These results suggest that light is necessary but insufficient to trigger chloroplast biogenesis in rice. Because light itself does not induce Chl accumulation in rice embryo (*Figure 1—figure supplement 2I–L*), we inferred that there are intrinsic cues to repress chloroplast development in rice embryo. However, to the best of our knowledge, the underlying mechanism that determines an embryo's ability to synthesize Chl is completely unknown. Here, we showed that OsNF-YB7, a LEC1 homolog of rice, acts as an inhibitor to repress Chl accumulation in the embryo. In line with this, a recent independent study also showed that the *OsNF-YB7* null mutant accumulates Chl in the embryo, although the cause of this remains unresolved (*Guo et al., 2022*).

Several TFs involved in chloroplast development and photomorphogenesis in rice have been identified (*Nakamura et al., 2009*; *Li et al., 2019*; *Bai et al., 2019*; *Hudson et al., 2013*). Interestingly, we found that many of them were upregulated in the *osnf-yb7* embryos. For example, *OsGLK1* was shown to be the most activated (*Figure 3A and B*). Our genetic and biochemical evidence suggests that *OsGLK1* is involved in the OsNF-YB7-mediated repression of Chl biosynthesis in rice embryo. *OsGLK1* overexpression mimicked the chloroembryo phenotype in WT, while knockout of *OsGLK1* and *OsGLK2*, simultaneously, suppressed Chl accumulation in *osnf-yb7* embryo (*Figure 4A–G*). The biochemical experiments suggested that OsNF-YB7 associates with the promoter of *OsGLK1*, in vivo, to transcriptionally inactivate *OsGLK1* (*Figure 3D–F*), indicating that *OsGLK1* is a downstream target of OsNF-YB7. However, OsNF-YB7 alone failed to bind to the promoter of *OsGLK1* in vitro (*Figure 3—figure supplement 1*), presumably due to some as-yet-unidentified TFs being recruited by OsNF-YB7, assisting in recognizing *OsGLK1* for transcriptional regulation. An NF-Y TF complex

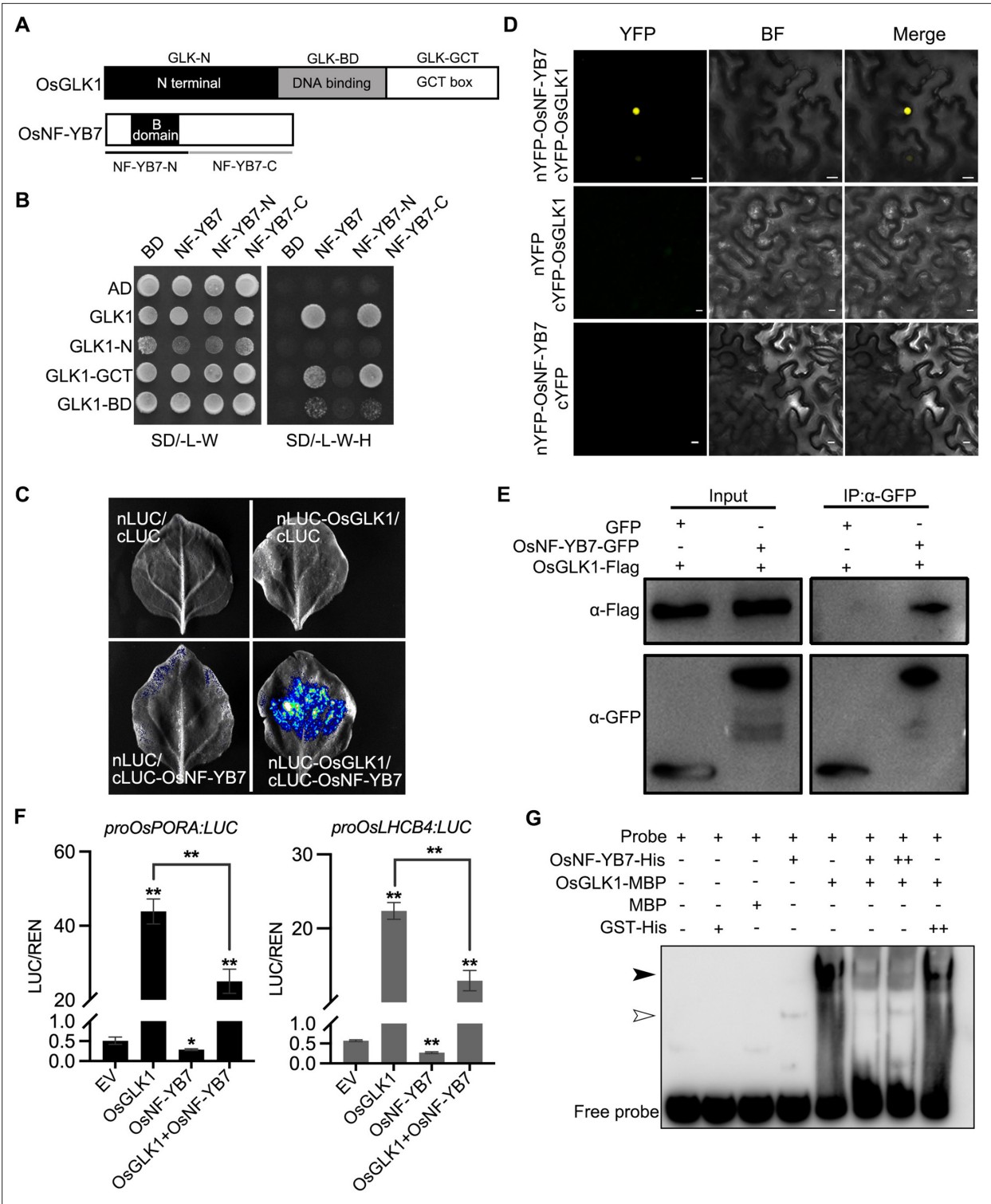

**Figure 6.** OsNF-YB7 interacts with OsGLK1 to regulate the expression of *OsPORA* and *OsLHCB4*. (**A**) Schematic diagrams showing the protein structures of OsGLK1 and OsNF-YB7. (**B**) Yeast-two-hybrid (Y2H) assays showing the interaction between OsNF-YB7 and OsGLK1. AD and BD indicate the activation domain and binding domain of GAL4, respectively. The full length or truncated OsNF-YB7 and OsGLK1 were fused with BD and AD, respectively. The indicated combinations of constructs were cotransformed into yeast cells and grown on the nonselective medium SD/-L-W and selective medium SD/-L-W-H. (**C**) A split complementary luciferase (LUC) assay confirmed the interaction between OsNF-YB7 and OsGLK1. Coexpression of the fusion of OsGLK1 and the N-terminal half of LUC (nLUC-OsGLK1) and the fusion of the C-terminal half of LUC and OsNF-YB7 (cLUC-OsNF-YB7) in the epidermal cells of *N. benthamiana* leaves induced LUC activities, whereas the epidermal cells coexpressing nLUC-OsGLK1 and

*Figure 6 continued on next page*

*Figure 6 continued*

cLUC, nLUC and cLUC-OsNF-YB7, or nLUC and cLUC did not show LUC activities. (**D**) Bimolecular fluorescence complementation (BiFC) assays showed interactions between OsNF-YB7 and OsGLK1 in the nucleus. OsNF-YB7 was fused with the N-terminal of yellow fluorescent protein (nYFP-OsNF-YB7); OsGLK1 was fused with the C-terminal of yellow fluorescent protein (cYFP-OsGLK1). The indicated combinations of constructs were transiently coexpressed in the leaf epidermal cells of *N. benthamiana*. Scale bar = 10 μm. (**E**) Co-immunoprecipitation (Co-IP) assays showing that OsNF-YB7 interacts with OsGLK1 in vivo. 35 S::OsNF-YB7:GFP (OsNF-YB7-GFP) and 35 S::OsGLK1:3×Flag (OsGLK1-Flag) were coexpressed in rice protoplasts and were immunoprecipitated with an anti-GFP antibody, and the immunoblots were probed with anti-GFP and anti-Flag antibodies. 35 S::GFP (GFP) was used as a negative control. (**F**) DLR assays showing that OsNF-YB7 represses the transactivation activity of OsGLK1 on *OsPORA* and *OsLHCB4*. Protoplasts isolated from etiolated seedlings were used for the analyses. EV, empty vector. Data are means ± SD (n=3). *, p<0.05; **, p<0.01; Student's *t*-test was used for statistical analysis. (**G**) EMSA assay indicated that OsNF-YB7 inhibits the DNA binding of OsGLK1 to the promoter of *OsPORA*. The black and white arrow heads indicate the OsGLK1- and OsNF-YB7-bound probes, respectively; "+" and "++" indicate that 2- and 4 μM recombinant proteins were used for the reactions. The GST-His was used as a negative control.

The online version of this article includes the following source data and figure supplement(s) for figure 6:

**Source data 1.** Uncropped and labelled gels.

**Source data 2.** Raw unedited gels.

**Figure supplement 1.** OsNF-YB7 interacts with OsGLK2.

**Figure supplement 2.** A proposed model of the OsNF-YB7-mediated suppression of Chl biosynthesis in rice embryo.

usually consists of three subunits (NF-YA/B/C) in order to exert its function (*Laloum et al., 2013*). NF-YA is responsible for DNA binding, while NF-YB and NF-YC are primarily responsible for transactivation (*Chaves-Sanjuan et al., 2021*; *Gnesutta et al., 2017*). OsNF-YB7 possibly forms a heterotrimer with NF-YA and NF-YC in the embryo to recognize *OsGLK1*. In addition, previous studies showed that *Arabidopsis* LEC1 can interact with different TFs for developmental regulation (*Huang et al., 2015b*; *Yamamoto et al., 2009*; *Fatihi et al., 2016*; *Boulard et al., 2018*; *Huang et al., 2015a*). Identifying such TFs may reveal how OsNF-YB7 recognizes *OsGLK1* in order to exert its function.

OsNF-YB7 could recognize a common set of genes involved in Chl biosynthesis and photosynthesis that are recognized by OsGLK1 (*Figure 2E–H* and *Figure 5—source data 1*). The biochemical results suggested that OsNF-YB7 can directly repress *OsPORA* and Os*LHCB4*, which are activated by OsGLK1 (*Figure 2J*, and *Figure 5—figure supplement 1*). OsNF-YB7 probably hinders OsGLK1 to access the target genes by forming an OsGLK1/OsNF-YB7 heterodimer (*Figure 6A–E*), or by occupying the motif OsGLK1 recognized in the promoter, given the binding sites of OsGLK1 and OsNF-YB7 are likely overlapped (*Figure 5F*, *Figure 5—figure supplement 3* and *Figure 5—source data 1*). The first of these hypotheses is more plausible because, upon co-incubation of OsGLK1 and OsNF-YB7 with the *OsPORA* promoter in vitro, the shifted band signal of OsGLK1 was substantially decreased (*Figure 6G*), indicating that the protein-protein interactions overwhelm the protein-DNA interactions. Thus, when OsNF-YB7 is expressed, it interacts with OsGLK1, then OsGLK1 is less available to activate downstream targets. The findings suggested that OsNF-YB7 plays a dual role in regulating Chl biosynthesis in rice embryo: first, it represses the downstream genes, achieved via its function as a transcriptional inactivator; second, OsNF-YB7 can interact with OsGLKs to disturb their abilities to transactivate genes related to Chl biosynthesis and photosynthesis (*Figure 6—figure supplement 2*). In addition to OsNF-YB7, a recent study showed that Deep Green Panicle 1, a plant-specific protein with a conserved TIGR01589 domain, can interact with OsGLKs to suppress OsGLK-mediated transcription (*Zhang et al., 2021b*). The findings suggested that, as the central regulator responsible for chloroplast development, GLK is tightly regulated at the post-translational level to fine-tune Chl biosynthesis in plants.

Previous studies suggested that *Arabidopsis* LEC1 is a positive regulator of Chl biosynthesis (*Pelletier et al., 2017*), given that the mature embryos of *lec1* were paler than the WT (*Meinke, 1992*; *West et al., 1994*). However, the null mutation of *OsNF-YB7* activated Chl biosynthesis, implying that the LEC1-type gene acts as a negative regulator in rice. By surveying the literature, we noted that Meinke reported that the cotyledons of the *lec1* mutant remained green unusually late in development (*Meinke, 1992*). Although there was no significant difference in Chl content upon using whole seeds for quantification, Parcy et al. did observe that the tip of *lec1* cotyledons accumulated more Chl (*Parcy et al., 1997*). Moreover, the *lec1;abi3* double mutant embryos produced much more Chl than the *abi3* single mutant (*Parcy et al., 1997*). Pelletier et al. recently reported that a cluster of LEC1 targets is enriched in genes related to photosynthesis and chloroplast development, and many

of which are down-regulated in the embryo of *lec1* (***Pelletier et al., 2017***); however, we noticed that in their dataset, the photosynthesis and Chl biosynthesis related genes were more strikingly enriched in the up-regulated genes of *lec1*, at either the mature green or postmature green stage. These observations challenge the concept that LEC1 positively regulates Chl biosynthesis and photosynthesis in *Arabidopsis*. Most studies in *Arabidopsis* have emphasized the importance of *LEC1* in embryo development at the maturation stage. However, *LEC1* is activated within 24 hr after fertilization (***Lotan et al., 1998***), but its role in the early embryo developmental stages for Chl biosynthesis is still unknown. Previously, we have expressed *OsNF-YB7* in the *lec1-1* background, driven by the native promoter of *Arabidopsis LEC1* (***Niu et al., 2021b***). Given that *OsNF-YB7* could rescue embryo morphogenesis defects in *Arabidopsis* (***Niu et al., 2021b***), we postulated that *OsNF-YB7* in rice plays a similar role to that of *LEC1* in *Arabidopsis*. In order to ascertain whether *LEC1* can fully restore *osnf-yb7*, it is necessary to ectopically express *LEC1* driven by the native *OsNF-YB7* promoter in the *osnf-yb7* background in the future, due to the possibility of functional divergence between the genes with regard to the regulation of chlorophyll biosynthesis in the embryo. It would also be worthwhile to carry out similar tests in other grass organisms to better understand the regulatory mechanism in rice.

In addition to producing chlorophyllous embryo, the *osnf-yb7* mutants display an array of developmental defects, including abnormal embryogenesis, reduced dormancy, and desiccation intolerance, similar to those found in *Arabidopsis lec1* mutants (***Niu et al., 2021a***). Mutation of *osglk1* and *osglk2* in *osnf-yb7* could recover the chloroembyo phenotype, but did not alleviate the other embryo defects (***Figure 4D–G***). The findings suggest that OsGLKs specifically function in Chl biosynthesis, but that OsNF-YB7 is responsible for many aspects of embryo development. In agreement with this, the number of OsNF-YB7-targeted genes is far greater than that of OsGLK-targeted genes, and the DEGs in *osnf-yb7* embryo is far greater than that in *OsGLK1-OX* embryo (***Figure 4H*** and ***Figure 5E***). The mechanisms underlying the OsNF-YB7-regulated multiple embryo developmental processes require further investigation.

# Materials and methods

## Plant materials and growth conditions

The *osnf-yb7* mutant lines used in this study were previously generated in our laboratory (***Niu et al., 2021b***). Rice cultivars Zhonghua11 (ZH11) and Kitaake (Kit) were used for gene transformation. Rice plants were grown in a paddy field in Yangzhou, Jiangsu Province, China, or in a climate-controlled room under long-day conditions with a photocycle of 14 hr of light (32 °C) and 10 hr of darkness (28 °C), at 50% humidity. The *N. benthamiana* plants were grown in a growth chamber at 22 °C with long-day conditions (16 hr light/8 hr dark). To determine whether the chloroembryo of *osnf-yb7* is light-dependent, the emerging panicles of the mutants were covered with aluminum foil until seed maturation. The lemmas of the WT were carefully removed with forceps at 1–2 DAF to expose the WT embryo to light in a climate-controlled room.

## Vector construction and plant transformation

To generate the overexpression plants, full-length coding sequences (CDSs) of *OsNF-YB7* or *OsGLK1* (the stop codon removed) were cloned into pCAMBIA1300-Flag or pUN1301-GFP under the control of *ubiquitin* promoter, using the ClonExpress II One Step Cloning Kit (Vazyme). The constructs were transformed into rice calli through an *Agrobacterium*-mediated strategy, as described previously (***Chen et al., 2016***). The higher order mutants of *OsNF-YB7*, *OsGLK1* and *OsGLK2* mutants, were generated using a previously described method for multiple gene editing in ZH11 (***Cheng et al., 2021***). The primers used for vector construction are listed in ***Supplementary file 1***.

## RNA extraction and RT-qPCR

Embryos or seedlings were collected by flash freezing in liquid nitrogen and stored at −80 °C until processing. Samples were finely ground using a mortar and pestle with liquid nitrogen. Total RNA was isolated using the RNA-easy Isolation Reagent (Vazyme, R701-01). The experiments were performed with at least three biological replicates. The relative expression levels of the tested genes were normalized using the rice *Actin* gene and calculated by the $2^{\Delta Ct}$ method. The primers used for the RT-qPCR are listed in ***Supplementary file 1***.

## RNA-sequencing and data analysis

Ten-DAF-old embryos of the Wild-type (WT) and *OsGLK1-OX* were used for RNA-sequencing. Two biological replicates were set. RNA extraction, library preparation, and high-throughput sequencing of the collected samples were outsourced to BGI Genomics Co., Ltd., Shenzhen, China. CLC Genomics Workbench 12.0 software was used for RNA-seq data analysis, as previously reported (*Xu et al., 2021*). The thresholds of fold change >2 and Bonferroni-corrected FDR <0.05 were used for defining a DEG. The previously generated ChIP-seq data of OsNF-YB7 and OsGLK1 (*Guo et al., 2022*; *Tu et al., 2022*) were reanalyzed using CLC Genomics Workbench 12.0 software for peak calling. Enriched motifs were identified by the online tool MEME-ChIP (https://meme-suite.org/meme/tools/meme-chip) with default parameter set. The software Mapman was used for pathway analysis (*Usadel et al., 2009*). The online tool AgriGO 2.0 was used for GO analysis (*Du et al., 2010*). The Venn diagrams were drawn using an online tool (https://bioinfogp.cnb.csic.es/tools/venny/index.html). TBtools was used for heat map generation (*Chen et al., 2020*).

## Dual luciferase reporter (DLR) assays

The *OsGLK1*, *OsLHCB4* (2 kb upstream of translation start site), and *OsPORA* (1.5 kb upstream of translation start site) promoter sequences were amplified from ZH11 genomic DNA and cloned into the vector pGreenII 0800-LUC (*Hellens et al., 2005*), as reporters; the *35 S::OsGLK1* and *35 S::Os-NF-YB7* constructs were used as effectors. The reporters and effectors were transfected into rice etiolated protoplasts in different combinations and incubated overnight. Firefly LUC and *Renilla* luciferase (REN) activities were measured using the DLR Assay Kit (Vazyme), following the manufacturer's instructions, and the LUC:REN ratios were calculated for analysis. The primers used for generating these constructs are listed in *Supplementary file 1*.

## Chromatin immunoprecipitation (ChIP) assays

The CDS (stop codons removed) of *OsGLK1* was cloned into the pJIT163-GFP vector driven by a 35 S promoter to generate the OsGLK1-GFP construct. Protoplasts transformed with OsGLK1-GFP and *OsNF-YB7-Flag* transgenic lines were used for the ChIP assays, in accordance with a previously described method (*Zhao et al., 2020*). Briefly, protoplasts or 0.2 g of 14-day-old seedlings were harvested and crosslinked with 1% formaldehyde for 15 min, followed by neutralization using 0.125 M glycine for an additional 5 min. The seedlings were then ground into powder in liquid nitrogen. The nuclei were isolated and lysed using Buffer S (50 mM HEPES-KOH [pH 7.5], 150 mM NaCl, 1 mM ethylenediaminetetraacetic acid [EDTA], 1% Triton X-100, 0.1% sodium deoxycholate, 1% SDS) and Buffer F (50 mM HEPES-KOH [pH 7.5], 150 mM NaCl, 1 mM EDTA, 1% Triton X-100, 0.1% sodium deoxycholate). The chromatin was then sonicated with the segment size ranging from 200 to 600 bp. The lysates were then immunoprecipitated by anti-GFP (abcam no. ab290) and anti-Flag (Sigma no. F3165) antibodies, respectively. Immunocomplexes were washed with low-salt ChIP buffer (50 mM HEPES-KOH, 150 mM NaCl, 1 mM EDTA, 1% Triton X-100, 0.1% sodium deoxycholate, 0.1% SDS), high-salt ChIP buffer (low-salt ChIP buffer but replacing 150 mM NaCl with 350 mM NaCl), ChIP wash buffer (10 mM Tris-HCl pH 8.0, 250 mM LiCl, 0.5% NP-40, 1 mM EDTA, 0.1% sodium deoxycholate), and TE buffer (10 mM Tris-HCl, pH 8.0, and 1 mM EDTA). The protein-DNA complexes were eluted from beads using ChIP Elution buffer (50 mM Tris-HCl pH 7.5, 10 mM EDTA, 1% SDS) for 15 min at 65 °C and the crosslinking was reversed by incubation overnight with proteinase K. The fragment DNA was extracted with phenol:chloroform:isoamyl alcohol (25:24:1), precipitated with ethanol, and resuspended in TE buffer. The immunoprecipitated DNA was used as a template for qPCR. The primers used here are listed in *Supplementary file 1*.

## Yeast two-hybrid assays

The CDSs of *OsGLK1/2* and *OsNF-YB7* were cloned into pGADT7 and pGBKT7, respectively. The constructs were cotransformed into yeast strain AH109 using Frozen-EZ Yeast Transformation II kit (Zymo), in accordance with the manufacturer's instructions. The empty pGADT7 and pGBKT7 vectors were cotransformed in parallel as negative controls. The transformants were first selected on synthetic dropout medium (SD/-Leu-Trp) plates. We tested protein-protein interactions using selective SD/-Leu-Trp-His dropout medium. Interactions were observed after 3 days of incubation at 28 °C. The primers used for generating these constructs are listed in *Supplementary file 1*.

## Split complementary LUC assays

Split complementary LUC assays were performed as previously described (*Niu et al., 2020*). The CDSs of *OsGLK1/2* and *OsNF-YB7* were cloned into JW771 and JW772 vectors to generate nLUC-OsGLK1/2 and cLUC-OsNF-YB7, respectively. The constructs were introduced into *Agrobacterium tumefaciens* strain GV3101 and then co-infiltrated into *N. benthamiana* leaves, and the LUC activities were analyzed after 48 hr of infiltration using Tanon Imaging System (5200 Multi; Tanon). The primers used for vector construction are shown in *Supplementary file 1*.

## Bimolecular fluorescence complementation assays

The CDSs of *OsNF*-YB7 and *OsGLK1* were cloned into the vector pSPYNE (nYFP) and pSPYCE (cYFP), respectively. The prepared plasmids were transformed into *Agrobacterium* strain GV3101, and the indicated transformant pairs were infiltrated into *N. benthamiana* leaves. Forty-eight hours after infiltration, the fluorescence signal of yellow fluorescent protein (YFP) was observed with a confocal laser scanning microscope (CLSM) (Carl Zeiss, LSM 710). Images were captured at 514 nm laser excitation and 519–620 nm emission for YFP. The primers used for vector construction are shown in *Supplementary file 1*.

## Co-immunoprecipitation (Co-IP) assays

Co-IP assays were performed using rice protoplast as described previously (*Zhang et al., 2011*). The CDSs (stop codons removed) of *OsGLK1* and *OsNF-YB7* were cloned into the vectors pUC19-35S-FLAG-RBS and pJIT163-GFP driven by a 35 S promoter, respectively. Ten micrograms of plasmid DNA (OsGLK1-GFP, GFP, and OsNF-YB7-Flag) was transformed or cotransformed into 200 μl of protoplasts and incubated in WI buffer (0.5 M mannitol, 20 mM KCl, and 4 mM MES at pH 5.7) for 12 hr. The protoplasts were collected and lysed in 500 μl of lysis buffer (50 mM Tris-HCl, 150 mM NaCl, 5 mM EDTA [pH 8.0], 1% NP-40, 0.1 mM PMSF). The extracts were incubated with GFP-Trap agarose beads at 4 °C for 3 hr and washed three times with washing buffer. Samples were boiled in SDS protein loading buffer. Immunoblots were detected using corresponding primary antibodies (anti-GFP, ABclonal no. AE012; anti-Flag, Sigma no. F3165). The primers used for vector construction are shown in *Supplementary file 1*.

## Chl measurement and confocal imaging

Approximately one hundred micrograms of embryos of the indicated genotypes were extracted in 1 ml of 100% dimethyl sulfoxide (DMSO) with incubation at 65 °C for 1 hr. Then, the absorbance values at wavelengths of 648.2 and 664.9 nm were measured by spectrophotometry and total Chl content was calculated as reported previously (*Barnes et al., 1992*). Chl autofluorescence signal was detected by CLSM (Carl Zeiss, LSM 710), with excitation and emission wavelengths of 633 and 625–730 nm, respectively.

## Electrophoretic mobility-shift assays (EMSAs)

*OsNF-YB7* CDS was amplified by PCR and cloned into pET-28a vector to generate the *OsNF-YB7-His* construct. The full-length CDS of *OsGLK1* was cloned into pMAL-c5X vector to generate *OsGLK1-MBP* construct. All constructs were transformed into *E. coli* strain BL21 to produce recombinant proteins. The promoter subfragments of *OsPORA* (42 bp, from –278—237) and *OsLHCB4* (38 bp, from –259—–222) were PCR amplified and labeled with biotin at the 3′ hydroxyl end of the double strands using EMSA Probe Biotin Labeling Kit (Beyotime, GS008). EMSA was performed using EMSA/Gel-Shift kit (Beyotime, GS009), in accordance with the manufacturer's instructions. The labeled probes were detected in accordance with the instructions provided with the EMSA/Gel-Shift kit. All oligonucleotides used to generate the biotin-labeled probes are listed in *Supplementary file 1*.

## Transmission electron microscopy (TEM)

TEM analysis was performed as described previously (*Cheng et al., 2021*). Briefly, embryos of WT and *osnf-yb7* were fixed overnight at 4 °C in 2.5% glutaraldehyde and 0.1 M PBS. The samples were subsequently washed three times with 0.1 M PBS and then fixed with 1% osmic acid for 4 hr. The samples

were dehydrated in a series of ethanol and embedded in acrylic resin at 37 °C for 12 hr and at 60 °C for 48 hr. The samples were sectioned at 100 nm and observed by TEM (TECNAI 12).

## Acknowledgements

We thank Prof. Hengxiu Yu and Dr. Chao Zhang of Yangzhou University for kindly providing the positive and negative control vectors for dual luciferase reporter assays. This research was supported by grants from the National Natural Science Foundation of China (32170344, 32300689), the Project of Zhongshan Biological Breeding Laboratory (BM2022008-02), the Jiangsu Province Government (JBGS001), the Natural Science Foundation of the Higher Education Institutions of Jiangsu Province (21KJB21003), the China Postdoctoral Science Foundation (2022M712702), the Independent Scientific Research Project Funds of the Jiangsu Key Laboratory of Crop Genomics and Molecular Breeding (PLR202101), the Priority Academic Program Development of Jiangsu Higher Education Institutions (PAPD).

---

## Additional information

### Funding

| Funder | Grant reference number | Author |
| --- | --- | --- |
| National Natural Science Foundation of China | 32170344 | Chen Chen |
| National Natural Science Foundation of China | 32300689 | Zongju Yang |
| Project of Zhongshan Biological Breeding Laboratory | BM2022008-02 | Chen Chen |
| Natural Science Foundation of the Higher Education Institutions of Jiangsu Province | 21KJB21003 | Zongju Yang |
| China Postdoctoral Science Foundation | 2022M712702 | Zongju Yang |
| Jiangsu Province Government | JBGS001 | Chen Chen |
| Priority Academic Program Development of Jiangsu Higher Education Institutions | PAPD | Chen Chen |
| Independent Scientific Research Project Funds of the Jiangsu Key Laboratory of Crop Genomics and Molecular Breeding | PLR202101 | Chen Chen |

The funders had no role in study design, data collection and interpretation, or the decision to submit the work for publication.

### Author contributions

Zongju Yang, Conceptualization, Data curation, Funding acquisition, Validation, Investigation, Visualization, Methodology, Writing - original draft; Tianqi Bai, Validation, Investigation, Visualization, Methodology; Zhiguo E, Software, Investigation; Baixiao Niu, Investigation; Chen Chen, Conceptualization, Supervision, Funding acquisition, Visualization, Writing - original draft, Project administration, Writing - review and editing

### Author ORCIDs

Zongju Yang ⓘ https://orcid.org/0000-0003-4700-049X

Chen Chen [ID] https://orcid.org/0000-0002-9748-3111

Reviewer #1 (Public review): https://doi.org/10.7554/eLife.96553.3.sa1
Reviewer #2 (Public review): https://doi.org/10.7554/eLife.96553.3.sa2
Reviewer #3 (Public review): https://doi.org/10.7554/eLife.96553.3.sa3
Author response https://doi.org/10.7554/eLife.96553.3.sa4

## Additional files

### Supplementary files
- Supplementary file 1. Primers used in this study.
- MDAR checklist

### Data availability
Sequencing data have been deposited in NCBI BioProject under the accession code PRJNA998591.

The following dataset was generated:

| Author(s) | Year | Dataset title | Dataset URL | Database and Identifier |
| --- | --- | --- | --- | --- |
| Yang Z, Niu B, Chen C | 2023 | OsNF-YB7 inactivates OsGLK1 to prevent chlorophyll biosynthesis in rice embryo | https://www.ncbi.nlm.nih.gov/bioproject/?term=PRJNA998591 | NCBI BioProject, PRJNA998591 |

The following previously published datasets were used:

| Author(s) | Year | Dataset title | Dataset URL | Database and Identifier |
| --- | --- | --- | --- | --- |
| Zhong S, Tu X | 2022 | GLK ChIP-seq and ATAC-seq data in Arabidopsis, Tobacco, Tomato, Rice and Maize | https://www.ncbi.nlm.nih.gov/geo/query/acc.cgi?acc=GSE220115 | NCBI Gene Expression Omnibus, GSE220115 |
| Guo F, Bian H, Zhang P, Wu Y | 2022 | ChIP-seq of OsLEC1 | https://www.ncbi.nlm.nih.gov/geo/query/acc.cgi?acc=GSE179596 | NCBI Gene Expression Omnibus, GSE179596 |

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
