## [Editor Report · eLife assessment]

This paper provides **important** insights into the role of rice OsNF-YB7, an ortholog of Arabidopsis LEC1, in chlorophyll biosynthesis, uncovering the genetic and molecular basis for negative regulation of chlorophyll production in the rice embryo. Mutational analysis, gene expression profiles and protein interaction combine for **convincing** evidence that OsNF-YB7 represses chlorophyll biosynthesis.

---

## [Referee Report · Reviewer #1 (Public review)]

Summary:

This manuscript investigates the regulation of chlorophyll biosynthesis in rice embryos, focusing on the role of OsNF-YB7. The rigorous experimental approach, combining genetic, biochemical, and molecular analyses, provides a robust foundation for these findings. The research achieves its objectives, offering new insights into chlorophyll biosynthesis regulation, with the results convincingly supporting the authors' conclusions.

Strengths:

The major strengths include the detailed experimental design and the findings regarding OsNF-YB7's inhibitory role.

Weaknesses:

However, the manuscript's discussion on the practical implications for agriculture and the evolutionary analysis of regulatory mechanisms could be expanded.

---

## [Referee Report · Reviewer #2 (Public review)]

Summary:

The authors set out to establish the role of the rice LEC1 homolog OsNF-YB7 in embryo development, especially as it pertains to the development of photosynthetic capacity, with chlorophyll production as a primary focus.

Strengths:

The results are well-supported and each approach used complements each other. There are no major questions left unanswered and the central hypothesis is addressed in every figure.

Weaknesses:

There are a handful of sections which could use clarifying for readers, but overall this is a solidly composed manuscript.

The authors clearly achieved their aims; the results compellingly establish a disparity between how this system operates in rice and Arabidopsis. Conclusions are thoroughly supported by the provided data and interpretations. This work will force a reconsideration of the value of Arabidopsis as a model organism for embryo chlorophyll biosynthesis and possibly photosynthesis during embryo maturation more broadly, as rice is a major crop organism and it very clearly does not follow the Arabidopsis model. It will thus be useful to carry out similar tests in other organisms rather than relying on Arabidopsis and attempt to more fully establish the regulatory mechanism in rice.

---

## [Referee Report · Reviewer #3 (Public review)]

Summary:

In this study, the authors set out to understand the mechanisms behind chlorophyll biosynthesis in rice, focusing in particular on the role of OsNF-YB7, an ortholog of Arabidopsis LEC1, which is a positive regulator of chlorophyll (Chl) biosynthesis in Arabidopsis. They showed that OsNF-YB7 loss-of-function mutants in rice have chlorophyll-rich embryos, in contrast to Arabidopsis LEC1 loss-of-function mutants. This contrasting phenotype led the authors to carry out extensive molecular studies on OsNF-YB7, including in vitro and in vivo protein interaction studies, gene expression profiling and protein-DNA interaction assays. The evidence provided well supported the core arguments of the authors, emphasising that OsNF-YB7 is a negative regulator of Chl biosynthesis in rice embryos by mediating the expression of OsGLK1, a transcription factor that regulates downstream Chl biosynthesis genes. In addition, they showed that OsNF-YB7 interacts with OsGLK1 to negatively regulate the expression of OsGLK1, demonstrating the broad involvement of OsNF-YB7 in rice Chl biosynthetic pathways.

Strengths:

This study clearly demonstrated how OsNF-YB7 regulates its downstream pathways using several in vitro and in vivo approaches. For example, gene expression analysis of OsNF-YB7 loss-of-function and gain-of-function mutants revealed the expression of selected downstream chl biosynthetic genes. This was further validated by EMSA on the gel. The authors also confirmed this using luciferase assays in rice protoplasts. These approaches were used again to show how the interaction of OsNF-YB7 and OsGLK1 regulates downstream genes. The main idea of this study is very well supported by the results and data.

Weaknesses:

It would be interesting to see how two similar genes have come to play opposite roles in Arabidopsis and rice. Interspecies complementation might help to understand this point.

---

## [Author Response]

The following is the authors’ response to the original reviews.

**eLife assessment**
This is an important study on the regulation of chlorophyll biosynthesis in rice embryos. It provides insights into the genetic and molecular interactions that underlie chlorophyll accumulation, highlighting the inhibition of OsGLK1 by OsNF-YB7 and the broader implications for understanding chloroplast development and seed maturation in angiosperms. The results presented, including mutation analysis, gene expression profiles, and protein interaction studies, provide convincing evidence for the function of OsNF-YB7 as a repressor in the chlorophyll biosynthesis pathway.

Thank you very much for your positive assessment of our manuscript. We have carefully revised the manuscript according to the reviewers’ valuable suggestions and comments. For more details, please see the point-to-point response to the reviewers below.

**Public Reviews:**

**Reviewer #1 (Public Review):**
Summary:This manuscript investigates the regulation of chlorophyll biosynthesis in rice embryos, focusing on the role of OsNF-YB7. The rigorous experimental approach, combining genetic, biochemical, and molecular analyses, provides a robust foundation for these findings. The research achieves its objectives, offering new insights into chlorophyll biosynthesis regulation, with the results convincingly supporting the authors' conclusions.Strengths:The major strengths include the detailed experimental design and the findings regarding OsNF-YB7's inhibitory role.Weaknesses:However, the manuscript's discussion on the practical implications for agriculture and the evolutionary analysis of regulatory mechanisms could be expanded.

Thank you for your insightful comments and suggestions. In the revised manuscript, we discussed the potential application of the chlorophyllous embryo (please see line 270-274). The presence of chlorophyll in the embryo facilitates photosynthesis at early developmental stages, potentially leading to improved seedling growth and vigor (Smolikova and Medvedev, 2016). In crops such as soybean and canola, green embryo is considered as a valuable trait due to its association with enhanced photosynthetic capacity, which consequently promotes fatty acid biosynthesis (Ruuska et al., 2004). However, chlorophyll degradation must be carefully managed during seed maturation to avoid negative effects on seed viability and meal quality (Chung et al., 2006). Interestingly, the green embryo of lotus (*Nelumbo nucifera*) is widely used as a food ingredient in Asian, Australia, and North America. It is employed in herbal medicine to treat nervous disorders, insomnia, and other conditions (Zhu et al., 2017; Ha et al., 2022), highlighting the significant potential value of the green embryo.

In many chloroembryophytes, such as Arabidopsis, the embryo occupies a large proportion of the seed. From an evolutionary perspective, the presence of chlorophyll in the embryo may promote adaptation in such chloroembryophytes because more reserves can be accumulated in the seed through active photosynthesis, better supporting the embryo development and subsequent seedling growth (Sela et al., 2020). On the other hand, some leucoembryophytes, such as rice, have persistent endosperm rich in storage reserves to nourish embryo development (Liu et al., 2022). Gaining the ability to accumulate chlorophyll in the embryo is unnecessary for such species. In agreement with this hypothesis, cholorophyllous embryos are more prevalent in non-endospermous seeds (Dahlgren, 1980). However, we would like to emphasize that the evolutionary force driving the divergence of chloroembryophytes and leucoembryophytes is currently almost completely unknown and deserves in-depth investigation in the future. We discussed the possible evolution of the ability to accumulate chlorophyll in the embryo, please find the details in Line 276-295.

**Reviewer #2 (Public Review):**
Summary:The authors set out to establish the role of the rice LEC1 homolog OsNF-YB7 in embryo development, especially as it pertains to the development of photosynthetic capacity, with chlorophyll production as a primary focus.Strengths:The results are well-supported and each approach used complements each other. There are no major questions left unanswered and the central hypothesis is addressed in every figure.Weaknesses:There are a handful of sections that could use clarifying for readers, but overall this is a solidly composed manuscript.The authors clearly achieved their aims; the results compellingly establish a disparity between how this system operates in rice and Arabidopsis. Conclusions are thoroughly supported by the provided data and interpretations. This work will force a reconsideration of the value of Arabidopsis as a model organism for embryo chlorophyll biosynthesis and possibly photosynthesis during embryo maturation more broadly, as rice is a major crop organism and it very clearly does not follow the Arabidopsis model. It will thus be useful to carry out similar tests in other organisms rather than relying on Arabidopsis and attempting to more fully establish the regulatory mechanism in rice.

Thank you very much for your positive comments. We have carefully revised the manuscript according to your and the other reviewers’ comments and suggestions. Particularly, we emphasized the necessary to carry out similar tests in other organisms rather than relying on Arabidopsis to better understand the regulatory mechanism in rice.

**Reviewer #3 (Public Review):**
Summary:In this study, the authors set out to understand the mechanisms behind chlorophyll biosynthesis in rice, focusing in particular on the role of OsNF-YB7, an ortholog of Arabidopsis LEC1, which is a positive regulator of chlorophyll (Chl) biosynthesis in Arabidopsis. They showed that OsNF-YB7 loss-of-function mutants in rice have chlorophyll-rich embryos, in contrast to Arabidopsis LEC1 loss-of-function mutants. This contrasting phenotype led the authors to carry out extensive molecular studies on OsNF-YB7, including in vitro and in vivo protein interaction studies, gene expression profiling, and protein-DNA interaction assays. The evidence provided well supported the core arguments of the authors, emphasising that OsNF-YB7 is a negative regulator of Chl biosynthesis in rice embryos by mediating the expression of OsGLK1, a transcription factor that regulates downstream Chl biosynthesis genes. In addition, they showed that OsNF-YB7 interacts with OsGLK1 to negatively regulate the expression of OsGLK1, demonstrating the broad involvement of OsNF-YB7 in rice Chl biosynthetic pathways.Strengths:This study clearly demonstrated how OsNF-YB7 regulates its downstream pathways using several in vitro and in vivo approaches. For example, gene expression analysis of OsNF-YB7 loss-of-function and gain-of-function mutants revealed the expression of selected downstream chl biosynthetic genes. This was further validated by EMSA on the gel. The authors also confirmed this using luciferase assays in rice protoplasts. These approaches were used again to show how the interaction of OsNF-YB7 and OsGLK1 regulates downstream genes. The main idea of this study is very well supported by the results and data.Weaknesses:From an evolutionary perspective, it is interesting to see how two similar genes have come to play opposite roles in Arabidopsis and rice. It would have been more interesting if the authors had carried out a cross-species analysis of AtLEC1 and OsNF-YB7. For example, overexpressing AtLEC1 in an osnf-yb7 mutant to see if the phenotype is restored or enhanced. Such an approach would help us understand how two similar proteins can play opposite roles in the same mechanism within their respective plant species.

We appreciate your insightful comments and suggestions. It is a very interesting question whether *AtLEC1* can fully restore *osnf-yb7*, given the possible functional divergence between the genes in terms of regulation of chlorophyll biosynthesis in the embryo. We have previously expressed *OsNF-YB7* in the *lec1-1* background in Arabidopsis, driven by the native promoter of *LEC1* (Niu et al., 2021). We found that *OsNF-YB7* could almost completely rescue the embryo defects in Arabidopsis, indicating that *OsNF-YB7* plays a resemble role in rice as the *LEC1* does in Arabidopsis (Niu et al., 2021). We sought to determine whether *AtLEC1* can complement the chlorophyll defect in *osnf-yb7*. However, given the fact that *osnf-yb7* shows severe callus induction defect, which is not surprising, because many studies have shown that *LEC1* is indispensable for somatic embryo development in various plant species, we are struggling to obtain the genetic materials for analysis. We have to transform *OsNF-YB7pro::AtLEC1* into the WT background first, and then cross the transformant with the *osnf-yb7* mutant. This is a time-consuming process in rice, but hopefully we will able to isolate a line expressing *OsNF-YB7pro::AtLEC1* in the *osnf-yb7* background from the resulting segregating population.

**Recommendations for the authors:**

**Reviewer #1 (Recommendations For The Authors):**
A minor comment regarding the chlorophyll contents quantification in the study. Line 87: "The results showed that WT had an achlorophyllous embryo throughout embryonic development,...." In the TEM result, chloroplast was not observed in the WT embryo sections, indicating a lack of chlorophyll-containing structures, contrary to what was found in the osnf-yb7 embryos where chloroplasts were observed.The authors stated that the embryo morphologies and Chl autofluorescence data showed that WT had an achlorophyllous embryo throughout embryonic development. However, the quantification of Chl levels in Figure 1D and Figure 4C showed that WT does produce some chlorophylls, albeit at lower levels than osnf-yb7 or OSGLK-OX embryos (WT values in the two figures are slightly different). This discrepancy warrants clarification to ensure consistency and accuracy in the manuscript's findings.

We re-evaluated the Chl content in the embryos of WT and *OsGLK1-OX* mature seeds. The result confirmed our previous finding that WT embryos produce a small amount of chlorophyll (please see the updated Fig. 4C). Notably, we observed that the dark-grown etiolated plants still have measurable chlorophyll content as reported in many studies (for example, Wang et al., 2017; Yoo et al., 2019), suggesting that there is potential bias in measuring chlorophyll content using an absorbance-based approach. We assume this possibly explains the concern you have raised.

**Reviewer #2 (Recommendations For The Authors):**
Mild editing for grammar is needed throughout, e.g. line 73, "It is still a mysterious why plant species".

We have carefully edited the grammar.

As a minor point, the placement of figure panels, such as in Figure 1, is not always intuitive.

Thank you for your suggestion. This figure has been revised as suggested. Please see the updated Fig. 1.

What is the significance of the two GFP mutants in Figures 2C and 2D? Is one of those the mislabeled Flag mutant?

The lines showed in Fig. 2C and D were not mislabeled. They were two independent transgenic events, both of which showed that OsNF-YB7 inhibited the expression of *OsPORA* and *OsLHCB4* in rice. The transgenic lines overexpressing *OsNF-YB7* tagging with the 3× Flag (*NF-YB7-Flag*) were also used for this experiment. In agreement, *OsPORA* and *OsLHCB4* were significantly downregulated in the three independent *NF-YB7-Flag* lines (Fig. S4C), confirming the results showed in Fig. 2C and D.

In Figures 2G and 2H, what is that enormous band at the bottom of the gel?

The bands at the bottom of the gel were free probes. We indicated this in the revised figure.

Not until the Materials and Methods section did I realize that any of this study was being done in tobacco; the Introduction implies it's rice vs. Arabidopsis and it might be a good idea to mention the organism of study somewhere before Figure 6.

We apologize for any confusion caused by our previous writing. While the majority of this study was performed with rice plants or protoplasts, the split complementary LUC assays and BiFC assays were performed with tobacco. We have specified these in the revised manuscript as suggested.

**Reviewer #3 (Recommendations For The Authors):**
It would be nice if the author could show what the phenotype is in AtLEC1 OX in osnf-yb7 and also OsNF-YB7 OX in atlec1 mutants.

Thank you for your suggestion. We have previously expressed *OsNF-YB7* in the *lec1-1* background of Arabidopsis, driven by the native promoter of Arabidopsis *LEC1* (Niu et al., 2021). Since *OsNF-YB7* could rescue the embryo morphogenesis defects in Arabidopsis (Niu et al., 2021), we assumed that *OsNF-YB7* plays a similar role in rice as the *LEC1* does in Arabidopsis. However, it remains unknown whether expression of *LEC1* in *osnf-yb7* may restore the chlorophyllous embryo phenotype in rice. As the generation of genetic material is time-consuming, and especially given the fact that *osnf-yb7* has a severe callus induction defect, we are struggling to obtain the complementary line for analysis. We have to transform *OsNF-YB7pro::AtLEC1* in a WT background first, and then cross the transformant with the *osnf-yb7* mutant. Hopefully, we will be able to isolate a line expressing *OsNF-YB7pro::AtLEC1* in *osnf-yb7* background, from the derived segregating population. We discussed the reviewer’s concern in the revised manuscript, please see Line 369-376.

Line 46, I think it is vague to mention that 'Like most plant species'. Some species might have different copy numbers, for example, a single GLK in liverwort M. polymorpha.

The statement has been revised. Please see Line 46.

Figures 2F and 5B, why was only one promoter region used for OsLHCB4? It would be better to have more regions like OsPORA.

Thank you for your comments. Here, we have examined more promoter regions (P1, P2 and P3) in the revised manuscript as suggested, among which, the previously selected promoter region (P3) contains both the G-box and CCAATC motifs that can be potentially recognized by GLK1. Consistent to our previous report, the results showed that OsNF-YB7 (left) and OsGLK1 (right) were associated with the P3 region, but showed no significant differences in the other probes. Please see the results in Fig. 2F and Fig. 5B of the revised manuscript.

Legend of Figures 2G, H, OsPORA (I), and OsLHCB (J) should be (G) and (H) respectively.

Corrected.

References

Chung, D.W., Pruzinska, A., Hortensteiner, S., and Ort, D.R. (2006). The role of pheophorbide a oxygenase expression and activity in the canola green seed problem. Plant Physiol 142, 88-97.

Ha, T., Kim, M.S., Kang, B., Kim, K., Hong, S.S., Kang, T., Woo, J., Han, K., Oh, U., Choi, C.W., and Hong, G.S. (2022). Lotus Seed Green Embryo Extract and a Purified Glycosyloxyflavone Constituent, Narcissoside, Activate TRPV1 Channels in Dorsal Root Ganglion Sensory Neurons. J Agric Food Chem 70, 3969-3978.

Liu, J., Wu, M.W., and Liu, C.M. (2022). Cereal Endosperms: Development and Storage Product Accumulation. Annu Rev Plant Biol 73, 255-291.

Niu, B., Zhang, Z., Zhang, J., Zhou, Y., and Chen, C. (2021). The rice LEC1-like transcription factor OsNF-YB9 interacts with SPK, an endosperm-specific sucrose synthase protein kinase, and functions in seed development. Plant J 106, 1233-1246.

Ruuska, S.A., Schwender, J., and Ohlrogge, J.B. (2004). The capacity of green oilseeds to utilize photosynthesis to drive biosynthetic processes. Plant Physiol 136, 2700-2709.

Sela, A., Piskurewicz, U., Megies, C., Mene-Saffrane, L., Finazzi, G., and Lopez-Molina, L. (2020). Embryonic Photosynthesis Affects Post-Germination Plant Growth. Plant Physiol 182, 2166-2181.

Smolikova, G.N., and Medvedev, S.S. (2016). Photosynthesis in the seeds of chloroembryophytes. Russ J Plant Physl+ 63, 1-12.

Wang, Z., Hong, X., Hu, K., Wang, Y., Wang, X., Du, S., Li, Y., Hu, D., Cheng, K., An, B., and Li, Y. (2017). Impaired Magnesium Protoporphyrin IX Methyltransferase (ChlM) Impedes Chlorophyll Synthesis and Plant Growth in Rice. Front Plant Sci 8, 1694.

Yoo, C.Y., Pasoreck, E.K., Wang, H., Cao, J., Blaha, G.M., Weigel, D., and Chen, M. (2019). Phytochrome activates the plastid-encoded RNA polymerase for chloroplast biogenesis via nucleus-to-plastid signaling. Nat Commun 10, 2629.

Zhu, M., Liu, T., Zhang, C., and Guo, M. (2017). Flavonoids of Lotus (Nelumbo nucifera) Seed Embryos and Their Antioxidant Potential. J Food Sci 82, 1834-1841.